# MTSSRL-MD: Multi-Task Self-Supervised Representation Learning for EEG Signals across Multiple Datasets

## Abstract

Electroencephalography (EEG) supports diverse clinical applications. However, effective EEG representation learning remains difficult because scarce label annotations and heterogeneous EEG montages limit the scale of available datasets. In practice, single and small-scale datasets often result in models with poor generalization, particularly for underrepresented classes with limited samples, which are harder to learn reliably. These challenges become even more critical in the EEG-based sleep stage classification task, especially for minority stages that are not only scarce but also transitional with overlapping characteristics, which makes them prone to misclassification. In this work, we propose *MTSSRL-MD (Multi-Task Self-Supervised Representation Learning for EEG Signals across Multiple Datasets)*, a unified framework that combines multi-dataset and multi-task self-supervised pretraining with a channel alignment module to alleviate the impact of scarce labels, heterogeneous EEG montages, and small-scale datasets that often cause poor generalization. This design enables the learning of EEG representations that are generalizable. Multi-dataset learning provides broader feature diversity that facilitates more robust cross-dataset generalization. A spatial-attention *Channel Alignment Module* (CAM) projects heterogeneous EEG montages into a shared channel space and provides spatial weights that highlight regions aligned with standard EEG montages, offering interpretability. Complementary self-supervised tasks—augmentation contrastive, temporal shuffling discrimination, and frequency band masking—provide temporal and spectral information that improve robustness on these underrepresented classes. Experiments on three heterogeneous EEG sleep datasets show that MTSSRL-MD consistently outperforms single-dataset SSRL baselines and even surpasses SeqCLR, a representative multi-dataset single-task SSRL method, particularly under low-label conditions, demonstrating the effectiveness of integrating multi-dataset and multi-task learning for EEG-based sleep stage classification. Besides classification performance, MTSSRL-MD achieves more efficient inference than single- and multi-dataset SSRL baselines. Moreover, the unified design of our proposed method allows the use of a single pretrained encoder to be fine-tuned across diverse datasets, highlighting efficiency and practical value for clinical research, suggesting strong potential for deployment in real-world settings.

## 1 Introduction

Electroencephalography (EEG) is a non-invasive modality with high temporal resolution, widely used in clinical and cognitive applications such as seizure detection, brain–computer interfaces, and sleep monitoring (Liu et al., 2025). Among these, **sleep stage classification** is particularly important for diagnosing disorders including insomnia, narcolepsy, and sleep apnea. (Zhang et al., 2024) Accurate identification of the five stages (W, N1, N2, N3, REM) defined by the American Sleep Disorders Association (AASM) is crucial for evaluating sleep architecture and treatment planning (Iber et al., 2007).

However, establishing a reliable sleep staging model remains challenging despite its clinical importance. Training deep learning models for biomedical applications typically requires large-scale

expert annotations, yet obtaining high-quality EEG labels is time-consuming and costly, as each recording demands long hours of manual review. Even trained annotators often disagree on transitional sleep stages (Lee et al., 2022; Reinke et al., 2025), further reducing the consistency of labeled data. As a result, the sample size of available labeled data is small, particularly for minority classes with limited samples, not only due to the sample size problem but also to transitional and overlapping characteristics, as mentioned before, that are harder to learn reliably. This makes supervised models susceptible to overfitting, leading to poor cross-dataset generalization, which limits the clinical applications. These challenges highlight the pressing need for approaches that can leverage large volumes of unlabeled EEG data. Self-supervised learning (SSL) has emerged as a promising solution, as it circumvents reliance on costly annotations by constructing supervisory signals directly from raw data. In this work, we specifically focus on self-supervised representation learning (SSRL), where the objective is to learn generalizable EEG representations from unlabeled recordings for downstream tasks (Eldele et al., 2023; Del Pup & Atzori, 2023; Weng et al., 2025).

Recently, multi-task self-supervised representation learning (MT-SSRL) methods have been developed. Compared with single-task SSRL, MT-SSRL introduces complementary pretext tasks to generate more diverse features. Since EEG data inherently contain rich temporal and spectral information, features obtained from a single task are often insufficient. Both temporal discrimination and frequency-related features are indispensable, making diverse pretext tasks particularly suitable for EEG and a common choice in recent studies. This technique helps the learned representations achieve better generalization (Li et al., 2022; Choi & Kang, 2024; Li et al., 2024a). Beyond addressing label scarcity through SSRL, like other biomedical modalities, EEG data share the same challenges, as they are inherently small in scale. Thus, representations learned solely from a single EEG dataset remain limited, motivating the use of multiple EEG datasets for SSRL. However, EEG datasets are inherently heterogeneous, as different hospitals use various devices that employ distinct electrode montages and varying channel counts, which hinders cross-dataset integration. Previous studies have explored multi-dataset SSRL strategies. SeqCLR (Mohsenvand et al., 2020) broadened representational diversity and improved generalization by training across multiple datasets. However, its reliance on a one-channel encoder discards inter-channel dependencies crucial for modeling spatial brain dynamics. Existing methods remain limited in their ability to capture complex brain activity patterns and to generalize across domains. No prior work unifies multi-dataset learning (distributional diversity) with multi-task SSRL (complementary temporal–spectral inductive biases) under effectively integrating multiple EEG datasets, leaving a gap for methods that improve cross-dataset generalization under scarce labels.

To address this gap, we propose *MTSSRL-MD (Multi-Task Self-Supervised Representation Learning for EEG Signals across Multiple Datasets)*, the first representation learning framework to combine **multi-dataset learning** and **multi-task self-supervised pretraining** for EEG, accounting for distributional diversity, complementary inductive biases, and montage alignment to tackle weak cross-dataset generalization caused by label scarcity, small sample sizes, and montage heterogeneity. Inspired by the spatial attention module (Défossez et al., 2023), we incorporate the *Channel Alignment Module* (CAM) that projects heterogeneous EEG montages into a shared channel space while preserving inter-channel dependencies, thereby harmonizing datasets and providing interpretable neurophysiological saliency, highlighting the importance of montage alignment. Three complementary SSRL tasks (augmentation contrastive learning, temporal shuffling discrimination, and frequency band masking) enrich temporal–spectral representations, enabling the model to learn more diverse and generalizable features. An adaptive uncertainty-weighted multi-task loss (Kendall et al., 2018) stabilizes joint optimization and automatically adjusts the contribution of each task during training. Evaluated on three heterogeneous public EEG sleep staging datasets (SleepEDF-20, ISRUC-S1, ANPHY-Sleep), MTSSRL-MD achieves state-of-the-art performance under low-label settings, achieving the highest Macro-F1-score gains, up to $+5.91\%$ over single-dataset SSRL baselines and $+11.37\%$ over SeqCLR at 5% labels, with corresponding gains of $+5.26\%$ and $+9.52\%$ at 10% labels, while remaining efficient, demonstrating *generalization*, *interpretability*, and strong potential for real-world deployment.

Finally, the remainder of this paper is organized as follows. Section 2 reviews related work on self-supervised representation learning and multi-dataset learning for EEG. Section 3 details the proposed MTSSRL-MD framework, covering the problem formulation, the overall framework overview, the pre-training stage with multi-dataset and multi-task design, and the fine-tuning stage. Section 4 describes the experimental setup and presents results, ablation studies, CAM visualiza-

tion, class-wise evaluation, and efficiency comparisons across three heterogeneous EEG datasets. Section 5 concludes the paper with a summary of contributions.

## 2 RELATED WORK

### 2.1 SELF-SUPERVISED REPRESENTATION LEARNING FOR EEG

Self-supervised representation learning (SSRL) has emerged as a promising approach for EEG analysis, as it alleviates reliance on costly annotations by constructing supervisory signals from raw recordings. (Banville et al., 2021) introduced influential single-dataset SSRL tasks such as Relative Positioning (RP), Temporal Shuffling (TS), and Contrastive Predictive Coding (CPC), showing that pretext tasks can extract meaningful temporal–spectral features. More recent works further validated SSRL in EEG-based sleep staging (Eldele et al., 2023), and (Li et al., 2024b) proposed occlusion-invariant objectives for medical time series such as EEG, confirming the effectiveness of SSRL in EEG and revealing strategies for improving temporal and spectral representations.

To further enrich learned features, multi-task SSRL (MT-SSRL) integrates complementary pretext tasks that capture diverse temporal–spectral features. Methods such as GMSS (Li et al., 2022) and MSLTE (Li et al., 2024a) combine contrastive, masking, and adaptive weighting. However, straightforward task combination often causes interference, motivating solutions such as heuristic weighting, gradient surgery, or uncertainty weighting (Kendall et al., 2018; Lin et al., 2021; Zhang & Yang, 2021). Among these, uncertainty-weighted loss has proven effective, as EEG SSRL studies (Li et al., 2022; 2024a) confirm its value for stabilizing training and improving representation quality by dynamically scaling the contribution of each task according to its predictive uncertainty. Despite these advances, most SSRL approaches remain confined to individual datasets, which restricts their ability to generalize across heterogeneous EEG datasets and limits robustness in cross-dataset scenarios.

### 2.2 MULTI-DATASET LEARNING FOR EEG

Leveraging multiple datasets is an effective strategy in vision and language for improving generalization (Muandet et al., 2013; Rebuffi et al., 2017). In EEG, however, cross-dataset training is considerably more challenging due to heterogeneity in electrode montages, recording devices, and sampling rates. Prior supervised work, such as RobustSleepNet (Guillot & Thorey, 2021), has shown that pooling datasets can improve robustness, highlighting the potential of multi-dataset integration for sleep staging. Several approaches have been proposed to mitigate montage differences. Channel-selection methods (Wei et al., 2022; Mellot et al., 2023) retain only overlapping electrodes but reduce spatial coverage. Feature-level fusion (Tveitstøl et al., 2024) aggregates dataset-specific representations, but no longer operates on aligned raw signals. Topology-aware and graph-based methods (Han et al., 2023; Yi et al., 2023) depend on dense and stable electrode layouts and therefore cannot be applied appropriately to low-density sleep EEG, where datasets include as few as two EEG channels (e.g., SleepEDF-20) and share no common montage. SeqCLR (Mohsenvand et al., 2020) avoids montage mismatch by using single-channel encoders, but sacrifices inter-channel dependencies that are important for modeling sleep-related rhythms.

Recent work on large-scale EEG pretraining, such as LaBraM (Jiang et al., 2024), CBraMod (Wang et al., 2024), and AdaBrain-Bench (Wu et al., 2025), aims to learn universal EEG representations across many datasets and tasks. These models provide valuable insights for broad EEG modeling, but they rely on large-scale datasets and substantial computational resources, and are optimized for general-purpose EEG transfer rather than for clinical sleep staging, where datasets are small, labels are limited, and fine-tuning must remain effective in low-label availability. These foundation-model approaches typically require substantial computational resources and large labeled datasets for downstream adaptation, which contradicts the constraints of our problem setting. Overall, existing approaches illustrate both the promise and the limitations of multi-dataset EEG learning. Without an explicit mechanism to harmonize heterogeneous channel layouts, models cannot fully leverage spatial information or scale reliably across datasets. This gap motivates the need for methods that explicitly align EEG montages and enable robust multi-dataset pretraining under realistic sleep-EEG conditions.

## 3 METHOD

### 3.1 PROBLEM FORMULATION

We aim to learn generalizable EEG representations from heterogeneous datasets with scarce labels. Formally, let $\mathcal{D} = \{D_1, \ldots, D_n\}$ denote a collection of $n$ heterogeneous EEG datasets, where each $D_i = \{(x_{ij}, y_{ij})\}_{j=1}^{N_i}$ with $x_{ij} \in \mathbb{R}^{c_i \times l}$ the $j$-th EEG segment ($c_i$ electrodes, length $l$) and $y_{ij}$ its label (when available).

Heterogeneity arises because each dataset follows its own montage ($c_i \neq c_j$), reflecting differences in channel count and electrode placement. To enable consistent learning, each input $x_{ij}$ is projected into a *unified channel space* by a dataset-specific Channel Alignment Module (CAM), which gives $u_{ij} = \mathcal{A}_i(x_{ij})$ where $\mathcal{A}_i : \mathbb{R}^{c_i \times l} \to \mathbb{R}^{c^* \times l}$.

A shared encoder $\mathcal{E} : \mathbb{R}^{c^* \times l} \to \mathbb{R}^d$ then transforms $u_{ij}$ into latent features $z_{ij} = \mathcal{E}(u_{ij})$. For downstream classification with $k$ sleep stages, a classifier $\mathcal{C} : \mathbb{R}^d \to \mathbb{R}^k$ maps $z_{ij}$ to label probabilities $P = \mathcal{C}(z_{ij})$.

### 3.2 FRAMEWORK OVERVIEW

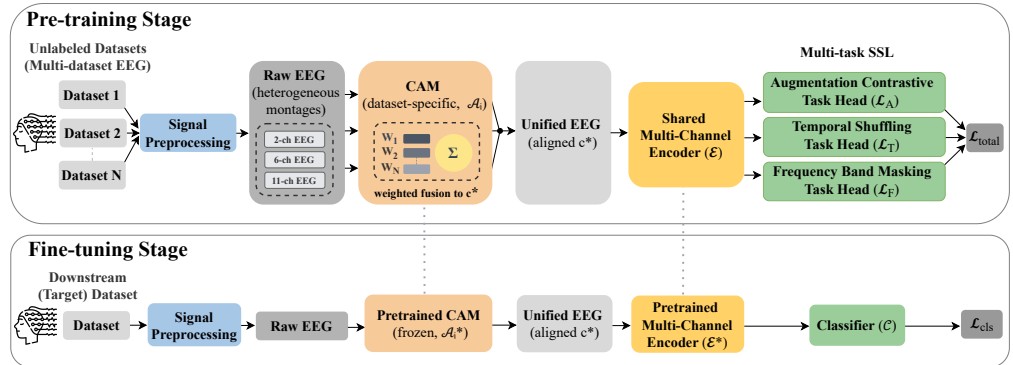

Figure 1: The overview of the proposed MTSSRL-MD for EEG representation learning. Stage 1 (**Pre-training**): dataset-specific CAMs align heterogeneous montages, and the shared encoder is trained with three SSRL tasks under uncertainty-weighted loss. Stage 2 (**Fine-tuning**): labeled data pass through the pretrained CAMs and encoder, and the classifier predicts sleep stage labels.

Our proposed framework **MTSSRL-MD** in Fig. 1 consists of two stages:

1. **Pre-training stage.** Unlabeled multi-dataset EEG segments are aligned by dataset-specific CAMs ($\mathcal{A}_i$) into a unified montage space, then passed through the shared encoder ($\mathcal{E}$). Three complementary SSRL tasks—augmentation contrastive, temporal shuffling discrimination, and frequency band masking—are optimized jointly with uncertainty-weighted loss $\mathcal{L}_{\text{total}}$.

2. **Fine-tuning stage.** Labeled data are processed by the pretrained CAMs (frozen, $\mathcal{A}_i^*$) and encoder ($\mathcal{E}^*$). A classifier $\mathcal{C}$ maps features to sleep stage labels, trained with cross-entropy loss $\mathcal{L}_{\text{cls}}$ under a progressive unfreezing schedule.

### 3.3 PRE-TRAINING STAGE: MULTI-DATASET LEARNING

#### 3.3.1 CHANNEL ALIGNMENT MODULE (CAM)

EEG datasets differ in channel count and montage, which hinders joint training. We modify the spatial attention module (Défossez et al., 2023) to construct our **Channel Alignment Module (CAM)** for multi-dataset EEG SSRL pretraining. Specifically, Fourier-based positional embeddings encode scalp geometry, and attention weights learn how to remap heterogeneous montages into a fixed

set of virtual channels $c^*$. Unlike the original work, which applied this mechanism for speech decoding within a single dataset, our CAM is adapted to the multi-dataset setting. Each dataset is equipped with a dataset-specific CAM, but all modules map to the same shared channel space, ensuring compatibility for subsequent joint training with the shared encoder. By balancing spatial integrity in low-channel recordings and redundancy in high-density montages, CAM serves as a unified geometry-informed alignment layer that enables stable cross-dataset training in MTSSRL-MD (detailed configuration is provided in Appendix G.1).

### 3.3.2 SHARED MULTI-CHANNEL ENCODER

After alignment, signals are processed by a shared encoder adapted from **DeepSleepNet** (Supratak et al., 2017), which uses dual-branch CNNs for temporal and spectral features. Unlike the original BiLSTM-based design, we replace recurrent layers with a multi-head self-attention module, enabling efficient long-range dependency modeling and better parallelization. Combined with CAM for channel alignment, the encoder produces temporal–spectral representations that remain consistent across heterogeneous datasets and transferable to downstream sleep staging.

## 3.4 PRE-TRAINING STAGE: MULTI-TASK SELF-SUPERVISED REPRESENTATION LEARNING

### 3.4.1 SELF-SUPERVISED REPRESENTATION LEARNING TASKS

We jointly optimize three complementary SSRL tasks, each intentionally designed to capture a different inductive bias relevant to sleep EEG—augmentation invariance, temporal continuity, and spectral discriminability. These three dimensions reflect the main types of variation commonly observed in sleep EEG signals. The first task, **augmentation contrastive task**, applies EEG-specific transformations (e.g., jitter, scaling, band-stop filtering) to generate two augmented views per segment, where the augmentation design follows **SeqCLR** (Mohsenvand et al., 2020) but the objective is optimized with the **SimSiam** framework (Chen & He, 2021), which encourages augmentation-invariant representations without requiring negatives (see Appendix E for detailed augmentation settings):

$$\mathcal{L}_A = -\tfrac{1}{2}\left[\cos(\tilde{\mathbf{p}}_1, \tilde{\mathbf{z}}_2) + \cos(\tilde{\mathbf{p}}_2, \tilde{\mathbf{z}}_1)\right], \tag{1}$$

where $\tilde{\mathbf{p}}_i = \mathbf{p}_i/\|\mathbf{p}_i\|$ and $\tilde{\mathbf{z}}_i = \mathbf{z}_i/\|\mathbf{z}_i\|$. To capture sequential dependencies, we adopt **temporal shuffling discrimination task** (Banville et al., 2021), where the model predicts whether subsequences preserve natural order or are shuffled, optimized with a Soft Margin loss,

$$\mathcal{L}_T = \tfrac{1}{N} \sum_{i=1}^{N} \log(1 + \exp(-y_i \cdot s_i)), \tag{2}$$

with labels $y_i \in \{-1, +1\}$ and score $s_i$. This objective encourages the encoder to capture the gradual evolution of sleep-related dynamics across time. This objective encourages the encoder to capture the gradual evolution of sleep-related dynamics across time. Finally, inspired by (Jo et al., 2023), we implemented **frequency band masking task** but adapted it to sleep staging with six canonical EEG bands (see Appendix E). In this task, one of the six bands (Delta–Gamma) is zeroed out in the Fourier domain, and the model is trained to classify the missing band using a cross-entropy loss.

$$\mathcal{L}_F = -\tfrac{1}{N} \sum_{i=1}^{N} \log p_\theta(c_i \mid \mathbf{z}_i), \tag{3}$$

with ground-truth index $c_i$ and softmax probability $p_\theta(c_i \mid \mathbf{z}_i)$. This encourages the encoder to learn discriminative spectral cues that are closely associated with sleep-related phenomena, such as delta activity in deep sleep, sigma spindles in N2, and increased theta rhythms during REM sleep. Together, these tasks offer complementary benefits. The augmentation contrastive task promotes robustness to signal and subject variability, the temporal shuffling task enhances sensitivity to how sleep-related patterns evolve over time, and the frequency band masking task strengthens discrimination of characteristic spectral features. Optimizing them jointly enables the encoder to incorporate these different aspects of sleep EEG into a unified representation.

### 3.4.2 MULTI-TASK OPTIMIZATION (UNCERTAINTY-WEIGHTED)

To balance the three SSRL tasks (augmentation contrastive $\mathcal{L}_A$, temporal shuffling discrimination $\mathcal{L}_T$, and frequency band masking $\mathcal{L}_F$), we adopt the **uncertainty-weighted loss** (Kendall et al., 2018). This approach introduces learnable homoscedastic uncertainties $\{\sigma_i\}$ that modulate the contribution of each task:

$$\mathcal{L}_{\text{total}} = \sum_{i \in \{A,T,F\}} \left( \frac{0.5}{\sigma_i^2} \, \mathcal{L}_i + \ln \sigma_i \right). \tag{4}$$

Tasks with lower uncertainty (small $\sigma_i$) are emphasized, while noisier or unstable tasks are down-weighted, avoiding manual tuning and reducing gradient interference. This adaptively balances the three tasks, achieving robust multi-task complementarity across heterogeneous datasets by integrating augmentation invariance, temporal dynamics, and frequency patterns into a unified representation. The empirical behavior of the learned uncertainty parameters and their resulting task contributions is reported in Appendix F.

### 3.5 FINE-TUNING STAGE

In the downstream task of sleep staging, labeled EEG segments are processed by the pretrained CAM and shared encoder to obtain feature representations. Within our proposed classifier, these features are aggregated using attention pooling and then mapped to class logits through an MLP layer. The classifier is trained using standard cross-entropy loss (full definition in Appendix D). To adapt pretrained features while avoiding catastrophic forgetting, we adopt a **progressive unfreezing schedule**, where initially only the classifier is updated, and encoder layers are gradually unfrozen. Importantly, **CAM** is kept **frozen** in evaluation mode, preserving the montage alignment learned during SSRL pretraining. This strategy ensures stable transfer across heterogeneous datasets and efficient generalization under both low- and full-label scenarios.

## 4 EXPERIMENT

### 4.1 EXPERIMENTAL SETUP

**Datasets.** We evaluate MTSSRL-MD on three public datasets: SleepEDF-20 (Kemp et al., 2000), ISRUC-S1 (Khalighi et al., 2016), and ANPHY-Sleep (Wei et al., 2024), all scored under the AASM five-stage standard (W, N1, N2, N3, REM) (Iber et al., 2007). These datasets exhibit cross-dataset heterogeneity in channel density (i.e., number of channels and montage configuration) and demographics. SleepEDF-20 (20 subjects, 2 channels) is relatively small and imbalanced; ISRUC-S1 (100 subjects, 6 channels) mixes healthy and disordered sleep; ANPHY-Sleep (29 subjects, 83 channels) is high-density but limited in scale. All data are resampled to 100 Hz, bandpass filtered (0.3–40 Hz), and segmented into 30s epochs. We adopt a subject-wise split to avoid subject leakage, consistent with prior works such as SeqCLR (Mohsenvand et al., 2020). All dataset statistics, electrode layouts, and data-split protocols are provided in Appendix A.

**Baselines.** We compare against: (1) supervised learning methods (DeepSleepNet (Supratak et al., 2017), U-Sleep (Perslev et al., 2021), ATCNet (Altaheri et al., 2022)); (2) single-dataset SSRL methods (RP, TS, CPC) (Banville et al., 2021); (3) multi-dataset SSRL method SeqCLR (Mohsenvand et al., 2020). All baselines follow the same training procedures under the same labeled-data settings. Performance is reported using Balanced Accuracy (BACC), Macro-F1-Score (MF1), and Cohen's Kappa ($\kappa$) as the main evaluation metrics. For the class-wise F1-Score analysis, we additionally report AUROC (AUC), as a threshold-independent separability measure. A complete description of hyperparameters and training configurations is reported in Appendix C.

### 4.2 RESULTS

### 4.2.1 OVERALL PERFORMANCE AND COMPARISON WITH BASELINES

**Single-dataset SSRL baselines.** Relative Positioning (RP), Temporal Shuffling (TS), and Contrastive Predictive Coding (CPC) (Banville et al., 2021) reduce label dependence by constructing pretext tasks within each dataset. However, these baselines are highly sensitive to the choice of

source and target datasets, and no single dataset consistently achieves the best transfer performance. This instability reflects a major drawback since single-dataset SSRL lacks robustness across domains and requires manual source selection, which limits its practicality for real-world deployment. By comparison, MTSSRL-MD achieves the highest observed gains, with relative improvements up to +5.37% in BACC and +5.91% in MF1 at 5% labels, and +5.26% in both metrics at 10% labels (Tables 1, 2). These results underscore that cross-dataset features are not only beneficial but often necessary for robust sleep staging. MTSSRL-MD addresses this challenge by explicitly aligning heterogeneous montages through CAM and jointly leveraging diverse temporal–spectral tasks, yielding stable and consistent improvements across datasets without requiring source-specific tuning.

Table 1: Downstream performance with 5% labeled fine-tuning across three datasets. Best results are in bold; second-best are underlined. Gray cells indicate in-domain settings.

| Pretrain Datasets | Models | SleepEDF-20 (S) | | | ISRUC-S1 (I) | | | ANPHY-Sleep (A) | | |
|---|---|---|---|---|---|---|---|---|---|---|
| | | BACC | MF1 | $\kappa$ | BACC | MF1 | $\kappa$ | BACC | MF1 | $\kappa$ |
| S | RP | 69.97± 1.81 | 67.13± 0.78 | 73.04± 1.18 | 60.30± 1.22 | 59.60± 1.01 | 53.11± 1.49 | 60.34± 1.89 | 60.03± 1.22 | 57.00± 1.95 |
| | TS | 68.69± 2.17 | 67.76± 0.90 | 74.70± 0.78 | 57.55± 1.20 | 57.68± 1.26 | 50.87± 2.01 | 60.13± 2.13 | 59.97± 1.58 | 56.43± 2.94 |
| | CPC | 72.23± 1.38 | 70.63± 0.86 | 77.95± 1.22 | 67.41± 0.77 | 67.15± 0.58 | 62.05± 1.28 | 63.48± 1.88 | 64.02± 1.78 | 62.11± 2.22 |
| I | RP | 65.78± 2.28 | 64.61± 1.41 | 72.36± 1.55 | 59.79± 0.53 | 59.31± 0.40 | 51.89± 1.36 | 60.86± 2.06 | 59.97± 1.34 | 55.21± 2.77 |
| | TS | 67.52± 1.65 | 65.40± 1.50 | 70.94± 2.38 | 59.40± 0.76 | 59.08± 0.18 | 52.10± 0.35 | 60.29± 2.37 | 59.58± 2.13 | 53.81± 3.85 |
| | CPC | 71.28± 1.44 | 69.73± 0.81 | 77.54± 0.86 | 66.21± 0.23 | 65.87± 0.29 | 60.93± 0.54 | 64.46± 1.74 | 64.53± 1.83 | 62.36± 2.04 |
| A | RP | 69.10± 1.66 | 67.39± 0.85 | 74.48± 0.85 | 62.87± 1.49 | 63.20± 1.34 | 57.67± 1.84 | 62.77± 0.81 | 63.35± 0.76 | 63.01± 0.93 |
| | TS | 69.60± 1.26 | 68.98± 0.81 | 77.00± 0.88 | 62.46± 1.71 | 62.05± 1.90 | 57.47± 2.43 | 64.28± 0.50 | 64.30± 1.00 | 64.09± 0.59 |
| | CPC | 69.06± 1.92 | 68.14± 1.01 | 75.26± 0.57 | 63.93± 0.65 | 64.07± 0.49 | 58.41± 1.06 | 64.73± 1.14 | 64.71± 0.99 | 64.08± 0.50 |
| S+I+A | SeqCLR | 71.02± 1.67 | 69.38± 0.82 | 76.21± 0.75 | 63.18± 0.22 | 63.86± 0.35 | 57.81± 1.75 | 64.31± 1.00 | 63.77± 1.26 | 62.50± 1.05 |
| | MTSSRL-MD | **76.07± 0.76** | **74.62± 0.90** | **80.53± 0.95** | **71.03± 0.63** | **71.12± 0.41** | **67.00± 0.75** | **68.02± 0.97** | **67.76± 0.84** | **66.04± 0.76** |

Table 2: Downstream performance with 10% labeled fine-tuning across three datasets. Best results are in bold; second-best are underlined. Gray cells indicate in-domain settings.

| Pretrain Datasets | Models | SleepEDF-20 (S) | | | ISRUC-S1 (I) | | | ANPHY-Sleep (A) | | |
|---|---|---|---|---|---|---|---|---|---|---|
| | | BACC | MF1 | $\kappa$ | BACC | MF1 | $\kappa$ | BACC | MF1 | $\kappa$ |
| S | RP | 70.90± 1.24 | 67.74± 0.64 | 73.83± 1.25 | 63.49± 1.68 | 62.76± 1.24 | 57.80± 2.02 | 62.68± 2.18 | 62.70± 1.99 | 59.84± 2.34 |
| | TS | 73.00± 1.50 | 70.07± 0.51 | 75.26± 1.28 | 67.14± 1.37 | 66.74± 1.44 | 62.37± 1.09 | 64.80± 1.17 | 62.17± 1.10 | 64.17± 1.09 |
| | CPC | 74.76± 0.38 | 72.07± 0.22 | 78.63± 0.49 | 68.45± 1.41 | 68.27± 1.30 | 64.02± 1.82 | 67.39± 1.37 | 67.60± 1.30 | 65.41± 1.55 |
| I | RP | 73.14± 1.02 | 70.17± 0.89 | 76.55± 1.10 | 67.37± 0.44 | 67.18± 0.48 | 62.33± 0.62 | 63.70± 1.51 | 63.30± 1.65 | 60.41± 2.57 |
| | TS | 73.10± 0.97 | 70.66± 0.46 | 76.60± 1.92 | 68.96± 0.78 | 68.86± 0.41 | 64.21± 0.71 | 65.59± 1.05 | 65.11± 1.19 | 60.16± 1.46 |
| | CPC | 75.56± 0.89 | 73.25± 0.52 | 79.70± 0.94 | 70.85± 0.30 | 70.53± 0.26 | 66.47± 0.60 | 69.35± 1.85 | 69.62± 1.44 | 67.10± 1.16 |
| A | RP | 72.73± 0.45 | 70.72± 0.32 | 77.78± 0.45 | 69.43± 0.33 | 68.83± 0.34 | 64.77± 0.35 | 66.74± 1.18 | 67.27± 1.26 | 65.42± 0.87 |
| | TS | 74.15± 1.06 | 72.02± 0.83 | 78.75± 0.76 | 70.74± 0.64 | 70.08± 0.49 | 66.44± 0.70 | 70.15± 0.47 | 70.26± 0.57 | 67.18± 1.35 |
| | CPC | 74.58± 0.90 | 71.84± 0.49 | 78.13± 1.61 | 70.74± 0.41 | 70.47± 0.26 | 66.41± 0.69 | 71.09± 0.97 | 70.91± 1.11 | 68.09± 1.14 |
| S+I+A | SeqCLR | 73.20± 1.35 | 70.25± 1.50 | 76.24± 1.04 | 68.26± 0.30 | 67.84± 0.38 | 63.94± 1.01 | 69.24± 1.16 | 68.80± 1.06 | 66.23± 1.26 |
| | MTSSRL-MD | **79.51± 0.79** | **76.94± 1.20** | **81.93± 0.95** | **74.61± 0.53** | **74.24± 0.35** | **68.93± 0.65** | **74.33± 0.93** | **74.52± 1.29** | **70.22± 0.80** |

Table 3: Downstream performance of MTSSRL-MD versus supervised learning baselines with 100% labeled fine-tuning across three datasets. Best results are in bold; second-best are underlined.

| Models | SleepEDF-20 (S) | | | ISRUC-S1 (I) | | | ANPHY-Sleep (A) | | |
|---|---|---|---|---|---|---|---|---|---|
| | BACC | MF1 | $\kappa$ | BACC | MF1 | $\kappa$ | BACC | MF1 | $\kappa$ |
| **Supervised Learning** | | | | | | | | | |
| DeepSleepNet | 77.54± 0.92 | 74.34± 0.81 | 79.83± 1.19 | 75.45± 0.30 | 74.54± 0.16 | 70.56± 0.18 | 78.90± 0.18 | 77.34± 0.43 | 75.90± 0.36 |
| U-Sleep | 79.95± 1.41 | 76.92± 0.97 | 82.15± 1.10 | 77.10± 0.73 | 77.06± 0.59 | 73.81± 0.35 | 77.20± 1.20 | 77.06± 0.50 | 74.72± 1.31 |
| ATCNet | 81.57± 0.84 | 78.46± 0.55 | 83.55± 0.73 | **78.14± 0.34** | **78.31± 0.15** | **74.71± 0.20** | 77.80± 1.10 | 77.60± 0.53 | 75.00± 0.52 |
| **Multi-Dataset Multi-Task SSRL** | | | | | | | | | |
| MTSSRL-MD | **82.26± 1.23** | **78.82± 0.64** | **83.96± 0.98** | 77.78± 0.26 | 77.83± 0.12 | 74.25± 0.12 | **79.26± 0.78** | **79.20± 0.64** | **77.15± 0.85** |

**Multi-dataset SSRL baselines.** MTSSRL-MD achieves the highest observed gains over Seq-CLR (Mohsenvand et al., 2020), with improvements of up to +12.42% in BACC and +11.37% in MF1 at 5% labels, and +9.30% (BACC) and +9.52% (MF1) at 10% labels (Tables 1, 2). Although SeqCLR demonstrates the benefit of dataset fusion, it still encounters essential limitations. Its single-channel encoder discards spatial dependencies across electrodes, limiting the use of multi-channel brain dynamics. This gap motivates explicit montage alignment, as addressed in MTSSRL-MD through the Channel Alignment Module (CAM), which harmonizes heterogeneous montages while

preserving inter-channel relationships. Moreover, SeqCLR employs only an augmentation-based objective, restricting robustness to temporal and spectral variations. MTSSRL-MD overcomes this by adopting a multi-task design that integrates augmentation contrastive, temporal shuffling discrimination, and frequency band masking tasks, which provide complementary inductive biases that enrich temporal–spectral representations. By combining montage alignment with multi-task objectives, MTSSRL-MD achieves stable and scalable generalization across heterogeneous EEG datasets.

**Supervised learning baselines.** With 100% label fine-tuning, MTSSRL-MD matches or outperforms strong supervised baselines (Table 3). On SleepEDF-20 and ANPHY-Sleep, MTSSRL-MD achieves the best results (82.26/78.82% BACC/MF1 and 79.26/79.20%), surpassing DeepSleep-Net (Supratak et al., 2017), U-Sleep (Perslev et al., 2021), and ATCNet (Altaheri et al., 2022). On ISRUC-S1, ATCNet achieves slightly higher performance, reflecting the clinical heterogeneity of the dataset, which allows specialized supervised models to leverage dataset-specific structures. However, MTSSRL-MD still performs competitively (77.78/77.83%, BACC/MF1) using a single pretrained model shared across datasets, demonstrating robustness under cross-dataset conditions. While supervised models can achieve strong performance with abundant labels, they tend to overfit to dataset-specific patterns and lack mechanisms for generalization across heterogeneous datasets. MTSSRL-MD also reaches strong clinical reliability, with $\kappa = 83.96\%$ (SleepEDF-20), 74.25% (ISRUC-S1), and 77.15% (ANPHY-Sleep).

Overall, MTSSRL-MD transforms dataset diversity into stable, label-efficient representations by combining *multi-dataset* pretraining with *multi-task* self-supervision. This design supports reliable generalization across heterogeneous sleep-EEG datasets, highlighting its strong potential for deployment in real-world settings.

### 4.2.2 Ablation and Analysis

To further validate the performance and effectiveness of our proposed MTSSRL-MD, we analyze multi-task weighting strategies, CAM weights, and class-wise F1-score analysis.

Table 4: Comparison of task weighting strategies in multi-task pretraining with 10% labeled fine-tuning. Best results are in bold; second-best are underlined.

| Strategies | SSRL Tasks | | | SleepEDF-20 (S) | | ISRUC-S1 (I) | | ANPHY-Sleep (A) | |
|---|---|---|---|---|---|---|---|---|---|
| | AUG | TS | FBM | BACC | MF1 | BACC | MF1 | BACC | MF1 |
| Equal weighting | ✓ | | | $77.34 \pm 0.45$ | $73.45 \pm 0.53$ | $72.25 \pm 0.27$ | $71.67 \pm 0.32$ | $72.61 \pm 1.47$ | $72.31 \pm 1.23$ |
| | | ✓ | | $77.82 \pm 0.87$ | $74.24 \pm 0.96$ | $\underline{73.79 \pm 0.22}$ | $\underline{73.34 \pm 0.16}$ | $72.20 \pm 1.07$ | $70.88 \pm 1.22$ |
| | | | ✓ | $\underline{78.23 \pm 0.79}$ | $\underline{75.61 \pm 0.68}$ | $73.51 \pm 0.13$ | $73.31 \pm 0.16$ | $\underline{74.50 \pm 1.28}$ | $\underline{73.94 \pm 1.04}$ |
| | ✓ | ✓ | ✓ | $77.55 \pm 0.77$ | $74.16 \pm 0.59$ | $72.58 \pm 0.86$ | $72.53 \pm 0.84$ | $73.43 \pm 1.50$ | $73.05 \pm 1.61$ |
| Uncertainty weighting | ✓ | ✓ | ✓ | $\mathbf{79.51 \pm 0.79}$ | $\mathbf{76.94 \pm 1.20}$ | $\mathbf{74.61 \pm 0.53}$ | $\mathbf{74.24 \pm 0.35}$ | $\mathbf{74.33 \pm 0.93}$ | $\mathbf{74.52 \pm 1.29}$ |

**Note:** Equal weighting denotes a fixed 1:1:1 ratio across tasks. Uncertainty weighting denotes the uncertainty-weighted loss. AUG refers to Augmentation Contrastive, TS to Temporal Shuffling Discrimination, and FBM to Frequency Band Masking.

**Comparison of Multi-Task Weighting Strategies.** As shown in Table 4, we compare equal and uncertainty weighting for combining augmentation contrastive, temporal shuffling discrimination, and frequency band masking pretext tasks. The results show that the utility of the task varies by dataset. Frequency patterns dominate in SleepEDF-20 and ANPHY-Sleep, whereas temporal patterns are critical for ISRUC-S1. The equal weighting strategy, which combines all three tasks with a fixed 1:1:1 ratio, does not fully exploit their complementary strengths and underperforms compared with the strongest single-task configuration. This gap reflects loss-scale imbalance and gradient interference. In contrast, uncertainty weighting consistently achieves the best result by adaptively down-weighting noisy objectives and emphasizing informative ones. Thus, MTSSRL-MD integrates conflicting objectives into complementary temporal–spectral invariances, enabling robust multi-task complementarity across heterogeneous datasets.

**Comparison of Multi-Dataset Setting.** Table 5 compares single-dataset pretraining (S-only / I-only / A-only) with multi-dataset pretraining (S+I+A (CAM)) under the same multi-task SSRL

Table 5: Ablation on multi-dataset pretraining with 5% and 10% labeled fine-tuning. Best results are in bold.

| Pretraining | Downstream Dataset | 5% BACC | 5% MF1 | 10% BACC | 10% MF1 |
|---|---|---|---|---|---|
| S-only | SleepEDF-20 (S) | 72.73±1.59 | 70.82±1.57 | 76.39±1.62 | 72.43±1.34 |
| S+I+A (CAM) | | **76.07±0.76** | **74.62±0.90** | **79.51±0.79** | **76.94±1.20** |
| I-only | ISRUC-S1 (I) | 68.98±1.94 | 68.52±1.11 | 72.04±0.78 | 71.78±1.34 |
| S+I+A (CAM) | | **71.03±0.63** | **71.12±0.41** | **74.61±0.53** | **74.24±0.35** |
| A-only | ANPHY-Sleep (A) | 64.32±1.70 | 63.12±1.37 | 70.60±2.08 | 70.03±0.52 |
| S+I+A (CAM) | | **68.02±0.97** | **67.76±0.84** | **74.33±0.93** | **74.52±1.29** |

objectives. Across all three downstream datasets, multi-dataset pretraining consistently outperforms single-dataset pretraining, achieving relative improvements of **+2.97 to +5.75% BACC** and **+3.79 to +7.35% MF1** under 5% labeled fine-tuning, and **+3.57 to +5.28% BACC** and **+3.43 to +6.41% MF1** under 10% labels. These improvements indicate that cross-dataset diversity provides complementary information that single-dataset learning cannot capture, and that CAM is crucial for harmonizing heterogeneous montages to effectively utilize this diversity. Combined with the task-level ablations in Table 4, the results show that MTSSRL-MD benefits from the combined effect of multi-task learning, dataset diversity, and explicit montage alignment.

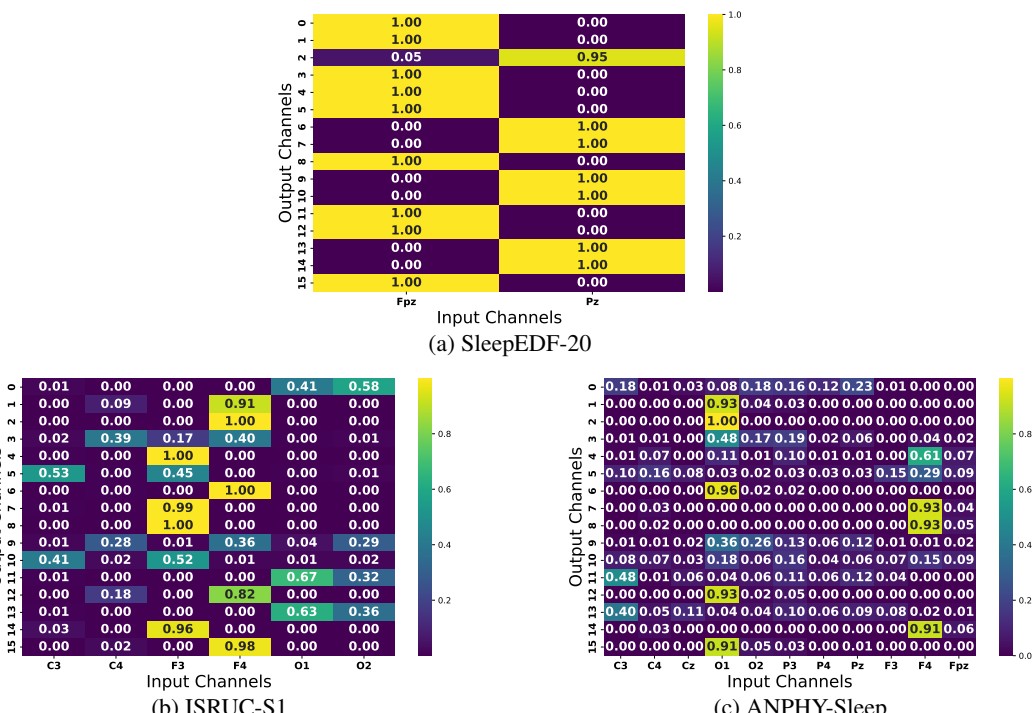

(a) SleepEDF-20

(b) ISRUC-S1

(c) ANPHY-Sleep

Figure 2: Spatial attention weights learned by Channel Alignment Module.

**CAM Weights Reveal Adaptive Montage Alignment.** Visualizations of CAM spatial attention (Fig. 2) show physiologically meaningful mappings: near-copy weights for SleepEDF-20 (2-channel), selective fusion and redundancy suppression for ISRUC-S1 (6-channel) and ANPHY-Sleep (11-channel). Without alignment, heterogeneous montages create input-level domain shift. CAM mitigates this by (i) harmonizing channels into a unified space, (ii) adaptively balancing sparse vs. dense montages, and (iii) highlighting channels that align with established neurophysiological findings. For example, CAM preserves Fpz–Pz in SleepEDF-20, emphasizes F3/F4 in ISRUC-S1,

and highlights O1/F4 in ANPHY-Sleep, consistent with known slow-wave and REM markers (Kemp et al., 2000; Khalighi et al., 2016; Wei et al., 2024; Frauscher et al., 2020). These unsupervised weighting patterns indicate that CAM not only facilitates cross-dataset alignment but also captures physiologically interpretable features. Further case analyses and clinical perspectives are provided in Appendix G.2.

Table 6: Overall and per-class performance on SleepEDF-20 with 10% labeled fine-tuning. Best results are in bold; second-best are underlined.

| Models | Overall Metrics (%) | | | | Per Class F1-Score (%) | | | | |
|---|---|---|---|---|---|---|---|---|---|
| | BACC | MF1 | AUC | $\kappa$ | W | N1 | N2 | N3 | REM |
| **Single-Dataset Single-Task SSRL** | | | | | | | | | |
| RP | 70.90± 1.24 | 67.74± 0.64 | 95.16± 0.54 | 73.83± 1.25 | 95.42± 0.46 | 21.26± 2.89 | 82.01± 1.35 | 83.47± 0.70 | 61.57± 1.03 |
| TS | 73.00± 1.50 | 70.07± 0.51 | 95.99± 0.28 | 75.26± 1.28 | 95.52± 0.61 | 28.03± 1.20 | 83.34± 0.75 | 84.30± 0.84 | 64.13± 1.65 |
| CPC | 74.76± 0.38 | 72.07± 0.22 | 96.34± 0.09 | 78.63± 0.49 | 96.93± 0.39 | 28.52± 2.35 | 84.36± 0.79 | 84.81± 1.09 | 68.74± 0.59 |
| **Multi-Dataset Single-Task SSRL** | | | | | | | | | |
| SeqCLR | 73.20± 1.35 | 70.25± 1.50 | 96.14± 0.36 | 76.24± 1.04 | **97.88± 0.84** | 32.07± 1.48 | 81.09± 1.32 | 82.45± 0.87 | 63.62± 1.03 |
| **Multi-Dataset Multi-Task SSRL** | | | | | | | | | |
| MTSSRL-MD | **79.51± 0.79** | **76.94± 1.20** | **97.44± 0.28** | **81.93± 0.95** | 97.26± 0.40 | **34.06± 2.52** | **86.83± 0.60** | **86.94± 0.65** | **72.93± 1.92** |

**Class-Wise F1-Score Analysis.** SleepEDF-20 is imbalanced, with N1 as a minority stage and REM characterized by ambiguous features that often overlap with other stages. With 10% labels, MTSSRL-MD achieves the largest gains on these challenging classes (N1: +6.2%, REM: +6.1% F1) while maintaining performance on majority stages. Naive dataset pooling alleviates N1 scarcity to some extent but provides limited benefit for REM, where SeqCLR even falls behind CPC. MTSSRL-MD delivers consistent improvements because its multi-task temporal–frequency objectives supply complementary information for disambiguating difficult stages, and CAM further aligns these patterns across datasets. Overall, combining multi-dataset diversity with multi-task objectives provides a more balanced and clinically reliable classifier, improving sensitivity to rare or ambiguous stages without sacrificing robustness on majority stages.

### 4.2.3 INFERENCE EFFICIENCY

MTSSRL-MD achieves the smallest computational footprint among all methods—only 0.01M parameters, 67 MB memory, and 3.2 ms latency per batch. Compared with SeqCLR, the multi-dataset SSRL baseline, MTSSRL-MD has $87\times$ fewer parameters (0.01M vs. 0.87M), uses $193\times$ less memory (67 MB vs. 12,948 MB), and reduces latency from 236.1 ms to 3.2 ms. Against supervised ATCNet, latency drops by $43\times$ (138.8 ms vs. 3.2 ms), while DeepSleepNet is even heavier. Although lightweight SSRL baselines (RP, TS, CPC) achieve comparable latency, they lack multi-dataset and multi-task capability. These results highlight MTSSRL-MD as the only approach that simultaneously delivers real-time efficiency and strong generalization across heterogeneous datasets. Complete efficiency comparisons are provided in Appendix H.

## 5 CONCLUSION

In this study, we propose **MTSSRL-MD** *(Multi-Task Self-Supervised Representation Learning across Multiple Datasets)*, a unified representation learning framework for EEG that combines **multi-dataset learning** and **multi-task self-supervised pretraining**. By integrating the Channel Alignment Module for montage alignment, three complementary SSRL objectives, and uncertainty-weighted optimization, MTSSRL-MD addresses weak cross-dataset generalization caused by label scarcity, small sample sizes, and montage heterogeneity. Experiments on three heterogeneous public datasets (SleepEDF-20, ISRUC-S1, ANPHY-Sleep) show consistent improvements over state-of-the-art baselines including SeqCLR (Mohsenvand et al., 2020), particularly in low-label settings. Besides, MTSSRL-MD improves the performance on difficult stages (N1, REM), learns physiologically meaningful spatial weights, and achieves computational efficiency, making it both interpretable and deployable. In summary, MTSSRL-MD enhances generalization, interpretability, and efficiency in a single framework, marking a step toward clinically practical and scalable EEG representation learning.

## REPRODUCIBILITY STATEMENT

We have taken several measures to ensure the reproducibility of our results. The architecture of MTSSRL-MD, including the Channel Alignment Module, shared encoder, and multi-task objectives, is described in detail in Section 3. Complete experimental settings, including dataset preprocessing, labeled data splits, hyperparameters, and training protocols, are reported in Section 4 and Appendix C. Evaluation metrics are formally defined in Appendix B, and dataset statistics with electrode layouts are provided in Appendix A. Together, these resources provide sufficient information for independent reproduction of our results. In addition, following the reproducibility guidelines, we will provide an anonymous repository link during the discussion phase to allow reviewers and area chairs to access our source code and reproduce all experiments in full. After this paper is accepted, we will provide the repository link in the abstract.

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

# A DATASET DETAILS

**Dataset Statistics and Electrode Configurations.** Table 7 reports complete statistics for the three datasets, including subject counts, channel configurations, and class distributions under the AASM five-stage standard. Figure 3 shows the electrode layouts, highlighting the heterogeneous montages that motivate the need for explicit alignment.

Table 7: Full statistics of the three EEG datasets, including class distributions under AASM five-stage standard.

| Datasets | #Subjects | Sampling Rate | #Channels | Class Distribution | | | | | |
| --- | --- | --- | --- | --- | --- | --- | --- | --- | --- |
| | | | | Wake | N1 | N2 | N3 | REM | #Total |
| SleepEDF–20 | 20 | 100 Hz | 2 | 72,354 (68.02%) | 2,804 (2.64%) | 17,799 (16.73%) | 5,703 (5.36%) | 7,717 (7.25%) | 106,377 |
| ISRUC–S1 | 100 | 200 Hz | 6 | 20,079 (23.06%) | 11,043 (12.68%) | 27,480 (31.55%) | 17,242 (19.80%) | 11,243 (12.91%) | 87,087 |
| ANPHY–Sleep | 29 | 1000 Hz | 11 | 3,322 (20.10%) | 1,663 (10.06%) | 6,564 (39.72%) | 2,985 (18.06%) | 1,992 (12.05%) | 16,526 |

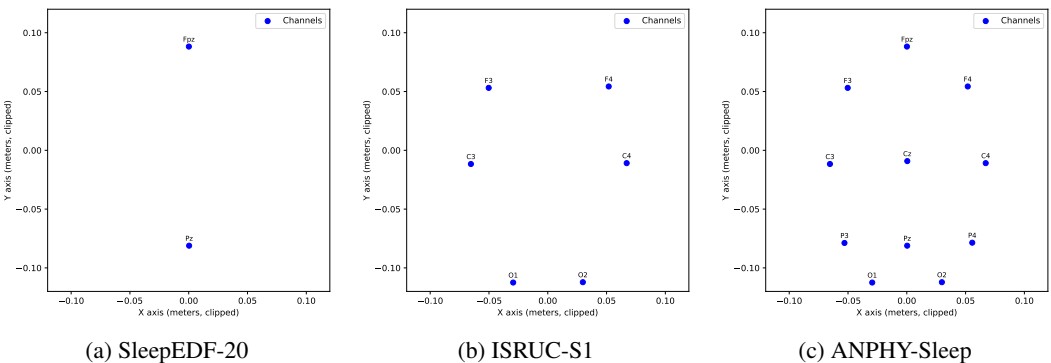

    (a) SleepEDF-20             (b) ISRUC-S1             (c) ANPHY-Sleep

Figure 3: Spatial layouts of EEG electrodes across datasets.

**Subject-wise Train/Validation/Test Split.** All experiments use strict subject-wise partitions to avoid subject leakage. For each dataset, 80% of subjects are assigned to the training split and 20% to the test split, and 20% of the training subjects are further held out as a validation set. The exact subject IDs for each split are included in the released configuration files.

**Definition of 5%, 10%, and 100% Labeled Fine-Tuning.** The subject-wise partitions remain fixed across all settings. The label-percentage conditions apply only within the training split. We perform stratified sampling to select 5%, 10%, or 100% of labeled epochs while maintaining class distribution. Validation and test sets always use all available labels.

# B EVALUATION METRICS

We adopt three primary metrics: Balanced Accuracy (BACC), Macro-F1-Score (MF1), and Cohen's $\kappa$. For the class-wise analysis, we additionally report AUROC (AUC) to complement per-class discriminability.

**Balanced Accuracy (BACC).**

$$\text{BACC} = \frac{1}{C} \sum_{c=1}^{C} \frac{\text{TP}_c}{\text{TP}_c + \text{FN}_c}, \tag{5}$$

which averages recall across all $C$ classes, mitigating the impact of class imbalance.

**Macro-F1-score (MF1).**

$$\text{MF1} = \frac{1}{C} \sum_{c=1}^{C} \frac{2 \cdot \text{Precision}_c \cdot \text{Recall}_c}{\text{Precision}_c + \text{Recall}_c}, \tag{6}$$

with $\text{Precision}_c = \frac{\text{TP}_c}{\text{TP}_c + \text{FP}_c}$ and $\text{Recall}_c = \frac{\text{TP}_c}{\text{TP}_c + \text{FN}_c}$.

**Cohen's $\kappa$.**

$$\kappa = \frac{p_o - p_e}{1 - p_e}, \tag{7}$$

$$p_o = \frac{1}{N} \sum_{i=1}^{N} \mathbf{1}(y_i = \hat{y}_i), \tag{8}$$

$$p_e = \sum_{c=1}^{C} \left( \frac{n_c}{N} \cdot \frac{m_c}{N} \right), \tag{9}$$

where $p_o$ is the observed agreement (accuracy) and $p_e$ is the expected agreement by chance, computed from label marginals with $n_c$ the number of true labels in class $c$ and $m_c$ the number of predicted labels in class $c$. Cohen's $\kappa$ accounts for chance agreement and is widely used in clinical EEG studies as a benchmark for inter-rater reliability.

**AUROC (AUC).** AUC is computed in a one-vs-rest manner for each class and then averaged across classes, providing a macro-level measure of discriminability. Unlike per-class F1, which relies on a fixed decision threshold, AUC evaluates the separability of score distributions across sleep stages. This provides a complementary perspective by assessing whether the model has learned to distinguish each stage from the others, even when class imbalance or suboptimal thresholds may suppress F1 performance.

## C  IMPLEMENTATION DETAILS

Experiments use PyTorch 2.2.2 with NVIDIA RTX 3090 GPUs. Encoders are pretrained with AdamW (400 epochs, batch size 100, learning rate $10^{-4}$), then fine-tuned with progressive unfreezing. CAM uses the cross-dataset default setting validated in our hyperparameter analysis (Appendix G.1).

## D  TRAINING OBJECTIVES

We pretrain with three SSRL objectives—augmentation contrastive learning (SimSiam objective), temporal shuffling discrimination (Soft Margin Loss), and frequency band masking (Cross Entropy Loss)—combined by an uncertainty-weighted multi-task loss. Formal definitions are provided in Section 3.4. For completeness, we restate only the fine-tuning objective below.

For downstream classification, we use the standard cross-entropy loss:

$$\mathcal{L}_{\text{cls}} = -\frac{1}{N} \sum_{i=1}^{N} \log p_\theta(y_i \mid \mathbf{x}_i), \tag{10}$$

where $p_\theta$ is the predicted probability of the true class label $y_i$.

## E  DATA AUGMENTATION AND FREQUENCY BANDS

Table 8 summarizes the EEG-specific transformations used in the augmentation contrastive task, and Table 9 lists the frequency bands adopted in the frequency band masking task.

**EEG Augmentations.** Following SeqCLR (Mohsenvand et al., 2020), we adopt six augmentation types, but adjust the parameter ranges to match the 30 s window length and 100 Hz sampling rate.

**EEG Frequency Bands.** For the frequency band masking task, we adopt six canonical EEG bands (Delta, Theta, Alpha, Sigma, Beta, Gamma), as defined in Table 9. Unlike (Jo et al., 2023), who used five bands, we explicitly separate Sigma (12–16 Hz) from Alpha, since Sigma activity corresponds to sleep spindles, an established biomarker of N2 sleep and a clinically relevant indicator of sleep stability and disorders (Memar & Faradji, 2017; Almutairi et al., 2023; Zaidi & Farooq, 2023).

Table 8: Transformation ranges used in data augmentation.

| Transformations | Min | Max |
|---|---|---|
| Amplitude scale | 0.5 | 2 |
| Time shift (samples) | $-100$ | $+100$ |
| DC shift ($\mu$V) | -10 | 10 |
| Zero-masking (samples) | 0 | 100 |
| Additive Gaussian noise ($\sigma$) | 0 | 0.2 |
| Band-stop filter (center frequency in Hz; bandwidth 2.5 Hz) | 2.8 | 47.5 |

Table 9: EEG frequency bands and their ranges.

| EEG Bands | Frequency Range (Hz) |
|---|---|
| Delta | 0.5–4 |
| Theta | 4–8 |
| Alpha | 8–12 |
| Sigma | 12–16 |
| Beta | 16–30 |
| Gamma | 30–50 |

# F ADDITIONAL ANALYSIS OF MULTI-TASK OPTIMIZATION (UNCERTAINTY WEIGHTING)

For completeness, we report the empirical behavior of the uncertainty-weighted objective (see Eq. (4) in the main paper). Following Kendall et al. (Kendall et al., 2018), each task is associated with a learnable homoscedastic variance $\sigma_i^2$, and its effective weight scales as $1/\sigma_i^2$. The learned variances converge to

$$\sigma_{\text{AUG}}^2 \approx 1 \times 10^{-4}, \qquad \sigma_{\text{TS}}^2 \approx 6.7, \qquad \sigma_{\text{FBM}}^2 \approx 1.34,$$

achieving the effective weight ratio

$$w_{\text{AUG}} : w_{\text{TS}} : w_{\text{FBM}} \approx 0.91 : 0.014 : 0.07.$$

To assess the contribution of each task to the final objective, we compute $w_i \cdot L_i$ at convergence:

$$w_{\text{AUG}} \cdot L_{\text{AUG}} \approx 1.48, \qquad w_{\text{TS}} \cdot L_{\text{TS}} \approx 0.50, \qquad w_{\text{FBM}} \cdot L_{\text{FBM}} \approx 0.50.$$

The three terms are within the same order of magnitude, demonstrating that uncertainty weighting normalizes the differences in raw loss scale and stabilizes multi-task training without optimization collapse or dominance by any individual task.

# G ADDITIONAL ANALYSIS OF CAM

## G.1 HYPER-PARAMETER ANALYSIS OF CAM

Fig. 4 and 5 summarize a grid search over two CAM hyper-parameters: the positional-embedding size (*Position Dimension*) and the number of virtual channels (*CM Channel*). We report downstream

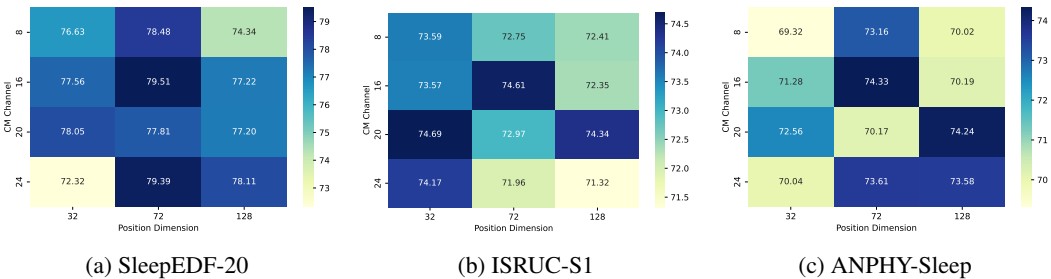

(a) SleepEDF-20          (b) ISRUC-S1          (c) ANPHY-Sleep

Figure 4: **BACC** heatmaps of CAM hyper-parameters on three datasets.

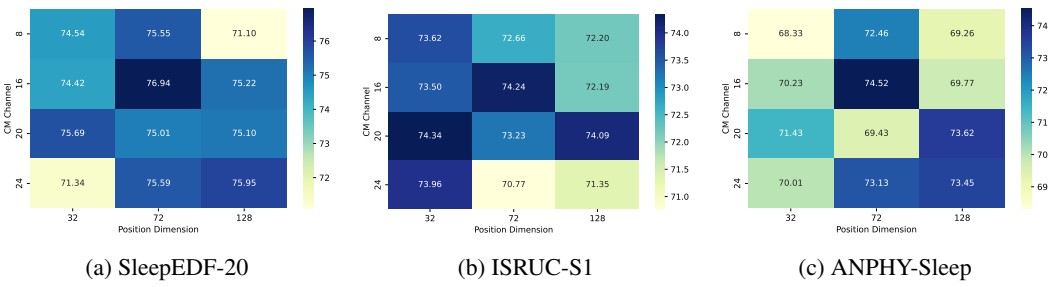

(a) SleepEDF-20          (b) ISRUC-S1          (c) ANPHY-Sleep

Figure 5: **MF1** heatmaps of CAM hyper-parameters on three datasets.

BACC and MF1 under 10% fine-tuning on SleepEDF-20 (S), ISRUC-S1 (I), and ANPHY-Sleep (A). Across S and A, the setting $(72, 16)$ yields the best results, while on I it ranks second with only a marginal gap, suggesting this configuration is consistently strong across the three datasets. Heatmaps further indicate why: very small embeddings underfit scalp geometry, overly large ones add little or destabilize training, too few channels discard spatial detail, and too many reintroduce redundancy. Overall, $(72, 16)$ achieves the best BACC–MF1 trade-off among tested configurations and represents a stable configuration.

## G.2 CAM WEIGHTS

Fig. 2 illustrates the spatial attention weights learned by CAM across the three datasets. Across settings, CAM does not assign uniform attention; instead, it adapts to each montage by reducing redundancy in higher-density configurations, preserving critical information in sparse ones, and highlighting channels consistent with established sleep biomarkers. These behaviors support both effective cross-dataset alignment and physiologically interpretable weighting.

**SleepEDF-20 Case Study.** On SleepEDF-20, CAM produces near-copy weights across the two available midline channels (Fpz–Pz), suggesting that in sparse montages, the module primarily preserves all available information. This finding is consistent with prior studies that have shown slow-wave dynamics relevant to sleep depth can be captured from midline and fronto-parietal derivations (Kemp et al., 2000).

**ISRUC-S1 Case Study.** ISRUC-S1 is a clinical cohort mixing healthy and disordered subjects (Khalighi et al., 2016), where REM sleep is often altered in terms of latency, density, or fragmentation. CAM consistently assigns higher weights to F3/F4 electrodes, aligning with the fronto-central distribution of REM sawtooth waves (Frauscher et al., 2020). This is particularly meaningful given that ISRUC includes both healthy and disordered populations, suggesting that CAM robustly emphasizes clinically relevant leads under heterogeneous conditions.

**ANPHY-Sleep Case Study.** On ANPHY-Sleep, CAM emphasizes O1 and F4, followed by C3 and O2/Pz, reflecting well-established sleep EEG generators. Occipital sites (O1/O2) as dominant sources of posterior alpha and visual-related activity (Halgren et al., 2019), fronto-central sites

(F4/C3) as key generators of slow waves (Massimini et al., 2004), sleep spindles (De Gennaro & Ferrara, 2003), and REM sawtooth waves (Frauscher et al., 2020), with activity extending toward midline parietal regions (e.g., Pz) (Nir et al., 2011). These correspondences suggest that CAM learns to exploit complementary spatial EEG sources when harmonizing heterogeneous high-density montages.

## H  INFERENCE EFFICIENCY

Table 10: Inference efficiency across supervised and SSRL models.

| Models | Parameters (M) | FLOPs (G) | Peak RAM (MB) | Latency (ms) |
|---|---|---|---|---|
| **Supervised Learning** | | | | |
| DeepSleepNet | 144.293 | 0.31 | 2710.5 | 4.0 |
| ATCNet | 0.113 | **0.03** | 94.6 | 138.8 |
| **Single-Dataset Single-Task SSRL** | | | | |
| RP | **0.007** | 0.10 | 84.0 | 3.9 |
| TS | **0.007** | 0.10 | 84.0 | 3.7 |
| CPC | **0.007** | 0.10 | 84.6 | 4.5 |
| **Multi-Dataset Single-Task SSRL** | | | | |
| SeqCLR | 0.868 | 0.31 | 12948.2 | 236.1 |
| **Multi-Dataset Multi-Task SSRL** | | | | |
| MTSSRL-MD | 0.010 | 4.87 | **67.0** | **3.2** |

**Note:** Parameters (M) denotes the number of parameters in millions. FLOPs (G) refers to the operations per forward pass. Latency denotes the mean single-batch forward time (batch size of 1). Peak RAM refers to the peak GPU memory usage recorded during the profiling run. Bold numbers indicate the best (lowest) results, and underlined numbers indicate the second-best within each column. All models are measured under identical input, precision, and hardware settings.

Table 10 reports parameters, FLOPs, memory footprint, and latency across baselines.

## I  FUTURE WORK

While MTSSRL-MD demonstrates strong generalization and efficiency, several directions remain open for exploration.

**Cross-task and cross-domain generalization.** Our pretraining is based on sleep staging dataset, which may bias features toward sleep-specific patterns. Extending to other EEG domains such as seizure detection, workload monitoring, or emotion recognition, could test generality. One concrete path is to curate a unified multi-domain benchmark and evaluate whether MTSSRL-MD achieves task-agnostic representations or requires domain-specific task heads.

**Adaptive channel alignment.** CAM currently learns dataset-level alignment, but EEG variability also exists at the subject/session level. Future work could explore adaptive or hierarchical CAMs that adjust spatial mappings at the subject or session level, supporting personalization in clinical applications.

**Semi-supervised and multimodal fusion.** Many clinical datasets include auxiliary modalities (EOG, EMG, ECG, clinical notes) or partially labeled data. A promising extension is to couple MTSSRL-MD with co-training or cross-modal contrastive objectives, enabling robust feature fusion and stronger performance when EEG-only labels are scarce.

