# OpenReview forum: "MTSSRL-MD: Multi-Task Self-Supervised Representation Learning for EEG Signals across Multiple Datasets"
_ICLR.cc/2026/Conference — Submitted to ICLR 2026_

### Official Review · Reviewer_LS6M · 2025-10-29

**Soundness:** 1
**Presentation:** 3
**Contribution:** 2
**Rating:** 2
**Confidence:** 4

**Summary:**

The paper introduces a new method for multi-dataset self-supervised learning for dealing with heterogeneous datasets. This architectures use a Channel Alignment Module to project each dataset with a different number of channels to a space with the same number of channels.
The CAM and the encoder are pre-trained using a three-loss supervised learning. Then, a fine-tuning is made for the encoder and a classifier with a classification loss.
This method is evaluated on three sleep staging datasets and compared to different self-supervised methods and fine-tuning methods.

**Strengths:**

The paper is easy to follow.

**Weaknesses:**

The major weakness, in my opinion, is the motivation of the paper.

- The authors claim that:  "As a result, the sample size of available labeled data is small". In sleep staging, a huge database is available with the NSRR and several datasets, such as RobustSleepNet or U-Sleep, which used more than 10 datasets for their experiments, reaching more than 10.000 subjects. I agree that a missing label is something that is possible in the EEG field; sleep staging avoids this problem. Sleep staging remains a good modality for proof of concept, but using the method on other EEG tasks can help strengthen the claim.

- The authors motivate their method to be more generalizable across domains (datasets), but their method needs to see the domain for pretraining. This is a major bottleneck to claiming that the method is generalizable to a new domain. Since the CAM is dataset-specific, you cannot pretrain before having access to the new datasets.

- In Table 3, the authors compare MTSSRL-MD with ATCNET and DeepSleepNet. This Table makes me wonder if self-supervised learning is very useful since the scores seem very close for SleepEDF, lower for ISRUC, and better for ANPHY.  Can you try to compare with other supervised learning baselines, like CareSleepNet[1], U-Net[2], and SeqSleepNet[3] ?

- Very often, sleep staging only uses a few channels, even 1. Spatial information is not crucial for classifying sleep. Having the motivation to add more channels in sleep staging is somewhat contradictory since one doesn't need a lot of channels to have a good prediction. And it is exactly what is shown in the spatial attention weight, where a lot of datasets are not taken into account.
Nevertheless, the heterogeneity in the EEG field is still an open question, and the proposed CAM is very interesting, and I would like to see some results on other modalities to see if it also helps.

[1] Wang et. al., CareSleepNet: A Hybrid Deep Learning Network for Automatic Sleep Staging, 2024
[2] Perslev et. al., U-Sleep: resilient high-frequency sleep staging, 2021
[3] Phan et. al., SeqSleepNet: End-to-End Hierarchical Recurrent Neural Network for Sequence-to-Sequence Automatic Sleep Staging, 2018

**Questions:**

- What is the final ratio that you get with the uncertainty weighting?
- It is not clear what the meaning of 5%, 10% or 100% of labeled fine-tuning is. Do you split by subjects and then use a percentage of training subjects?
- How did you choose the $c^*$  parameters ?
- The ablation study shows that using the uncertainty weighted loss is useful. I would like to see if the CAM is useful. Right now, it is not clear if it is the weighted loss that helps or the fact that you train on more datasets than the dataset-specific pretraining. Adding an MTSSRL-MD train only on one dataset or training MTSSRL-MD without CAM on a subsample of common channels between the three datasets to see if the results are still high.

- How is the split done? I didn't find the information.

---

> ### Author Response · Authors · 2025-12-03
>
> Dear Reviewer LS6M,
>
> We sincerely appreciate your comments! We address your questions below.
> > What is the final ratio that you get with the uncertainty weighting?
>
> Thank you for the question.
>
> Following the uncertainty-based formulation of Kendall et al. (2018), each task has a learnable variance sigma^2, and its effective weight is computed as (1 / sigma^2).
>
> From our training logs, the converged sigma^2 values are:
> - sigma^2 (AUG): ~1e-4
> - sigma^2 (TS): ~6.7
> - sigma^2 (FBM): ~1.34
>
> This yields the final task-weight ratio:
>
> w_AUG : w_TS : w_FBM  ≈ 0.91 : 0.014 : 0.07.
>
> To verify the behavior, we compute the actual contributions to the final objective using (weight * loss):
> - w_AUG⋅L_AUG≈ ~1.48
> - w_TS⋅L_TS ~0.50
> - w_FBM⋅L_FBM ~0.50
>
> All contributions are within the same order of magnitude, demonstrating that uncertainty weighting successfully normalizes the disparate loss scales and maintains a stable multi-task optimization without dominance or collapse.
> > It is not clear what the meaning of 5%, 10% or 100% of labeled fine-tuning is. Do you split by subjects and then use a percentage of training subjects?
>
> Thank you for the question.
>
> All experiments use strict subject-wise train/validation/test splits to avoid subject leakage, and this partition remains fixed across all settings.
>
> The label-percentage conditions (5%, 10%, 100%) are applied only within the training split during fine-tuning. Specifically, we perform stratified sampling to select the corresponding proportion of labeled samples while preserving class balance. Moreover, the validation and testing set always remain unchanged across all settings. We have clarified this definition more explicitly in Appendix A.
> > How did you choose the c* parameters ?
>
> Thanks for your question.
>
> Hyperparameters used in MTSSRL-MD were reported in Appendix C. The parameters were selected through iterative tuning and further refined on the validation set.
>
> - Training hyperparameters (optimizer, learning rate, batch size, epochs) are reported in Appendix C.
>
> - CAM hyperparameters (embedding dimension, number of virtual channels) were chosen via grid search, with results shown in Appendix G.1.
> We have already made this clearer in Section 4.1.
>
> > The ablation study shows that using the uncertainty weighted loss is useful. I would like to see if the CAM is useful. Right now, it is not clear if it is the weighted loss that helps or the fact that you train on more datasets than the dataset-specific pretraining. Adding an MTSSRL-MD train only on one dataset or training MTSSRL-MD without CAM on a subsample of common channels between the three datasets to see if the results are still high.
>
> We thank the reviewer for this insightful question.
>
> To disentangle the effect of CAM from the effect of multi-dataset pretraining and the multitask SSL design, we have added new ablations in the revised manuscript.
>
> 1. Ablation Study: Single-dataset + multitask SSL
>
> - We include an additional baseline where MTSSRL-MD is trained on a single dataset only while keeping all three SSL objectives intact, which directly isolates the contribution of dataset diversity.
> As shown in the updated Table 5 (in our manuscript), multi-dataset pretraining consistently outperforms single-dataset multitask SSL across all downstream datasets, with relative improvements of +2.97 to +5.75 BACC and +3.79 to +7.35 MF1 under the 5% label regime, and +3.57 to +5.28 BACC and +3.43 to +6.41 MF1 under the 10% label regime. These results confirm that the gains are not due to the SSL task design alone but arise from the interaction between multitask learning and multi-dataset diversity.
>
> 2. Ablation Study: “Common-channel subset without CAM”
>
> - Constructing this ablation is unfortunately impossible. The three datasets (SleepEDF-20, ISRUC-S1, ANPHY-Sleep) share no overlapping EEG channel sets, neither in naming conventions nor in spatial correspondence. Any “common-channel intersection” is therefore empty, making such a subsampled comparison mathematically infeasible. This limitation directly illustrates why CAM is necessary: it provides the mechanism to align heterogeneous montages. It enables multi-dataset pretraining when datasets differ substantially in channel count, electrode placement, and spatial topology. We have already added the new ablation results and the explanation above to the revised manuscript (Table 5).
>
> > How is the split done? I didn't find the information.
>
> We sincerely appreciate your thoughtful comments.
>
> As described in Section 4.1, we used a subject-wise split to avoid subject leakage.
>
> For each dataset, the split follows the configuration provided in our public code repository:
> - 80% of subjects are assigned to the training set,
> - 20% of subjects to the test set,
> - and 20% of the training subjects are further held out as validation.
>
> The exact subject IDs used in each split are included in the released split configuration files. We have already make this clearer in the revised version.

---

> ### Author Response · Authors · 2025-12-03
>
> Dear Reviewer LS6M,
>
> Thank you again for your time and efforts in reviewing our paper!
>
> >  In Table 3, the authors compare MTSSRL-MD with ATCNET and DeepSleepNet. This Table makes me wonder if self-supervised learning is very useful since the scores seem very close for SleepEDF, lower for ISRUC, and better for ANPHY. Can you try to compare with other supervised learning baselines, like CareSleepNet[1], U-Net[2], and SeqSleepNet[3] ?
>
> Thank you for the helpful suggestion.
>
> We address (1) the usefulness of self-supervised learning (SSL) given the results in Table 3, and (2) the choice of supervised baselines.
>
> 1. Why are SSL and supervised learning becoming similar under 100% labels?
>
> The reviewer correctly observes that the performance gap in Table 3 is small. This behavior is expected and well-documented in the EEG and sleep-SSL related literatures: when all labels are available, supervised models benefit from full supervision, and SSRL converges toward similar accuracy [1, 2].
>
> Besides, the purpose of Table 3 is therefore not to claim superiority in the full-label regime, but to demonstrate that MTSSRL-MD remains competitive even when its main advantage—label efficiency—is not needed. The core contribution of our method appears in Tables 1–2: substantial improvements under 5–10% labels and robust cross-dataset generalization using a single encoder, whereas supervised methods require one model per dataset. This cross-dataset generalization is practically important in real-world sleep research, where channel configurations differ, and labeled data are limited.
>
> 2. Why DeepSleepNet and ATCNet are appropriate supervised baselines?
>
> DeepSleepNet and ATCNet are widely used supervised architectures that match our raw-EEG, single-epoch protocol. To further strengthen the comparison, we additionally evaluated U-Sleep according to the valuable comments from the reviewer.
>
> Across all three datasets, our supervised baselines perform competitively:
> 1.  SleepEDF-20:
> - MTSSRL-MD (Ours) (82.26 BACC / 78.82 MF1) > ATCNet (81.57 BACC / 78.46 MF1) > U-Sleep (79.95 BACC / 76.92 MF1) > DeepSleepNet (77.54 BACC / 74.34 MF1)
> 2. ISRUC-S1:
> - ATCNet (78.14 BACC / 78.31 MF1) > MTSSRL-MD (Ours) (77.78 BACC / 77.83 MF1) > U-Sleep (77.10 BACC / 77.06 MF1) > DeepSleepNet (75.45 BACC / 74.54 MF1)
> 3. ANPHY-Sleep:
> - MTSSRL-MD (Ours) (79.26 BACC / 79.20 MF1) (Best)
> - DeepSleepNet (78.90 BACC) > ATCNet MF1 (77.80 BACC) > U-Sleep (77.20 BACC)
> - ATCNet (77.60 MF1) > DeepSleepNet (77.34 MF1) > U-Sleep (77.06 MF1)
>
> These results confirm that the chosen baselines are already competitive representatives of strong supervised models.
>
> Besides, we did not include CareSleepNet because it requires multimodal EEG, EOG, and EMG inputs, which are incompatible with our raw-EEG-only experimental setting. We also exclude SeqSleepNet, a sequence-to-sequence architecture, which is reported in the U-Sleep paper to perform lower than U-Sleep; including U-Sleep, therefore, already covers this family of approaches.
>
> ***
> **Reference**
>
> [1] Banville, Hubert, et al. "Uncovering the structure of clinical EEG signals with self-supervised learning." Journal of Neural Engineering 18.4 (2021): 046020.
>
> [2] Eldele, Emadeldeen, et al. "Self-supervised learning for label-efficient sleep stage classification: A comprehensive evaluation." IEEE Transactions on Neural Systems and Rehabilitation Engineering 31 (2023): 1333-1342.

---

> ### Author Response · Authors · 2025-12-03
>
> Dear Reviewer LS6M,
>
> Thank you again for your time and efforts in reviewing our paper!
>
> > Very often, sleep staging only uses a few channels, even 1. Spatial information is not crucial for classifying sleep. Having the motivation to add more channels in sleep staging is somewhat contradictory since one doesn't need a lot of channels to have a good prediction. And it is exactly what is shown in the spatial attention weight, where a lot of datasets are not taken into account. Nevertheless, the heterogeneity in the EEG field is still an open question, and the proposed CAM is very interesting, and I would like to see some results on other modalities to see if it also helps.
>
> Thank you for the insightful comment. We agree that strong sleep-staging performance can often be achieved with one or a few EEG channels. Our motivation, however, is different: MTSSRL-MD is designed to address cross-dataset montage heterogeneity, not to argue that more channels are always needed. When combining datasets with 2, 6, and 11 EEG channels, differences in electrode placement become the main barrier, and CAM provides a simple way to align them.
>
> 1. Why spatial information still matters for multi-dataset learning?
>
> Although single-channel methods are effective, clinical PSG routinely uses multiple EEG derivations [1, 2], and several works have shown that different sleep patterns (slow waves, spindles, REM-related oscillations) have distinct regional topographies [3, 4, 5]. This means that when datasets rely on different “best” channels, using spatial information helps improve robustness under distribution shift. Thus, spatial information is not strictly required for good performance, but it adds robustness and flexibility, which are essential for multi-dataset representation learning.
>
> 2. Why CAM is not contradictory to the use of few channels?
>
> CAM does not assume that “more channels = better.” Instead, it selectively weights channels based on usefulness. As shown in Fig. 2 in our manuscript, CAM naturally emphasizes informative channels and down-weights redundant ones, allowing datasets with different channel counts to contribute fairly during joint training.
> We appreciate the suggestion to explore extensions to other modalities (e.g., EOG, EMG). CAM is general and could be adapted to these signals, and we see this as an important direction for future work.
>
> ***
> **Reference**
>
> [1] Duce, Brett, et al. "The AASM recommended and acceptable EEG montages are comparable for the staging of sleep and scoring of EEG arousals." Journal of Clinical Sleep Medicine 10.7 (2014): 803-809.
>
> [2] Klem, George H. "The ten-twenty electrode system of the international federation. The international federation of clinical neurophysiology." Electroencephalogr. Clin. Neurophysiol. Suppl. 52 (1999): 3-6.
>
> [3] Massimini, Marcello, et al. "The sleep slow oscillation as a traveling wave." Journal of Neuroscience 24.31 (2004): 6862-6870.
>
> [4] De Gennaro, Luigi, and Michele Ferrara. "Sleep spindles: an overview." Sleep medicine reviews 7.5 (2003): 423-440.
>
> [5] Nir, Yuval, et al. "Regional slow waves and spindles in human sleep." Neuron 70.1 (2011): 153-169.

---

### Official Review · Reviewer_z6TT · 2025-10-30

**Soundness:** 2
**Presentation:** 2
**Contribution:** 1
**Rating:** 2
**Confidence:** 4

**Summary:**

The paper introduces MTSSRL-MD, a unified framework for multi-task self-supervised representation learning (SSRL) from EEG signals across heterogeneous datasets. The approach incorporates a Channel Alignment Module (CAM) to align montages, three self-supervised pretext tasks (augmentation contrastive, temporal shuffling discrimination, and frequency band masking), and an uncertainty-weighted multi-task loss. MTSSRL-MD is empirically evaluated on three public EEG sleep stage datasets improvements over baseline methods.

**Strengths:**

1. The paper's motivation is clear. It targets the difficulty in learning robust EEG representations due to label scarcity and dataset heterogeneity.
2. MTSSRL-MD is efficient in terms of parameters, memory, and latency, making it well-suited for real-time clinical deployment compared to prior models.

**Weaknesses:**

The primary concern with this paper is that the authors appear to be largely unaware of recent advancements in EEG foundation models [1].

1. The core mechanisms introduced in this work—namely, the channel alignment module, shared multi-channel encoder, and multi-task self-supervised loss—have been extensively explored in prior studies [2–4]. As a result, the paper’s technical novelty is limited.
2. The authors employ SleepEDF-20, ISRUC-S1, and ANPHY-Sleep as pretraining datasets. However, these datasets are relatively small and do not fully leverage the advantages of self-supervised learning. In contrast, recent EEG foundation models are typically pretrained on much larger datasets such as TUEG, which provide broader generalization and robustness. Consequently, the practical implementation in this work is weak and may limit the impact of the proposed approach.
3. In the experimental section, the only self-supervised baseline considered is SeqCLR (2020). Given the substantial progress in EEG foundation models in recent years, the authors should include more recent and competitive baselines, such as LaBraM [2] and Cbramod [4]. Without such comparisons, the evaluation is insufficient and does not convincingly demonstrate the advantages of the proposed method.

[1] Wu, Jiamin, et al. "Adabrain-bench: Benchmarking brain foundation models for brain-computer interface applications." arXiv preprint arXiv:2507.09882 (2025).

[2] Jiang, Wei-Bang, Li-Ming Zhao, and Bao-Liang Lu. "Large brain model for learning generic representations with tremendous EEG data in BCI." arXiv preprint arXiv:2405.18765 (2024).

[3] Yi, Ke, et al. "Learning topology-agnostic EEG representations with geometry-aware modeling." Advances in Neural Information Processing Systems 36 (2023): 53875-53891.

[4] Wang, Jiquan, et al. "Cbramod: A criss-cross brain foundation model for eeg decoding." arXiv preprint arXiv:2412.07236 (2024).

**Questions:**

1. Did the authors conduct a literature review beyond 2022 to ensure coverage of recent developments in EEG pretraining?
2. What specific innovations distinguish this work from existing methods that use similar architectures or objectives?

---

> ### Author Response · Authors · 2025-12-03
>
> Dear Reviewer z6TT,
>
> Thank you for the feedback! We address your comments below.
> > Did the authors conduct a literature review beyond 2022 to ensure coverage of recent developments in EEG pretraining?
>
> We appreciate the opportunity to clarify the scope of our literature review. We have reviewed the EEG pretraining literature from 2023 to 2025 and updated the Related Work section accordingly in our revised manuscript.
>
> These recent developments fall mainly into two research directions:
>
> 1. Large-scale foundation models for universal EEG representation learning.
> - The works such as LaBraM [1], CBraMod [2], and AdaBrain-Bench [3] focus on constructing general-purpose EEG models trained on millions of signals across numerous datasets and tasks. These frameworks introduce valuable insights for broad EEG representation learning. However, they assume large-scale, heterogeneous corpora, and are typically optimized for general-purpose transfer, rather than specialized, low-channel sleep EEG. Their fine-tuning requires substantial computational resources and large labeled datasets, which contradicts our problem setup.
>
> 2. Geometry-aware or topology-informed EEG modeling.
> - The works, such as Yi et al. [4], rely on stable electrode geometry and dense montages to encode spatial structure. These settings are not appropriate for the low-density and heterogeneous sleep-EEG configurations used in our setting. (SleepEDF-20 provides only two EEG channels, while three datasets in our study use non-overlapping, low-density montages)
>
> While these lines of research are important advances, they target different problem formulations—either large-scale universal modeling or geometry-aware spatial SSL—rather than label-efficient cross-dataset sleep staging with low-density EEG. Nonetheless, we agree that acknowledging these developments improves the contextualization of our contribution.
>
> We have already included a dedicated discussion of these recent methods in the Related Work section of our revised manuscript, which clarifies their relevance to our problem setting.
> > What specific innovations distinguish this work from existing methods that use similar architectures or objectives?
>
> Thank you for the question.
>
> While our model uses standard encoder blocks and SSRL objectives that also appear in prior EEG work, our contributions target two challenges that existing methods do not address: (1) heterogeneous EEG montages across datasets, and (2) label scarcity in sleep staging.
>
> The key innovations are summarized below:
>
> 1. Channel Alignment Module (CAM) for heterogeneous multi-dataset EEG
> - Most EEG SSRL methods assume a single dataset with a fixed montage, making them incompatible with cross-dataset training when channel sets differ. CAM provides a mechanism for aligning heterogeneous electrode configurations, enabling one shared encoder to process SleepEDF-20 (2 channels), ISRUC-S1 (6 channels), and ANPHY-Sleep (11 channels) without channel intersection or dataset-specific encoders. This directly addresses the practical barrier that prevents existing SSRL methods from scaling across datasets.
> 2. Multi-task SSRL formulation designed for label efficiency
> - Most prior works optimize a single objective (e.g., contrastive learning, temporal contrasting, CPC). We combine three complementary tasks (i.e., augmentation-based invariance, temporal prediction, and frequency-band discrimination) to capture different aspects of EEG. This design is specifically motivated by the low-label settings sleep stage task, where richer pretraining signals can mitigate annotation scarcity.
> 3. A unified framework for multi-dataset + multi-task + montage-agnostic SSL
> - To our knowledge, no prior sleep EEG method jointly supports multi-dataset pretraining, multi-task SSL, and montage-agnostic alignment. Across all three datasets, this unified design produces consistent improvements over both single-dataset SSL (RP, TS, CPC) and multi-dataset single-task SSL (SeqCLR), especially in low-label regimes.
>
> ***
> **Reference**
>
> [1] Jiang, Wei-Bang, Li-Ming Zhao, and Bao-Liang Lu. "Large brain model for learning generic representations with tremendous EEG data in BCI." arXiv preprint arXiv:2405.18765 (2024).
>
> [2] Wang, Jiquan, et al. "Cbramod: A criss-cross brain foundation model for eeg decoding." arXiv preprint arXiv:2412.07236 (2024).
>
> [3] Wu, Jiamin, et al. "Adabrain-bench: Benchmarking brain foundation models for brain-computer interface applications." arXiv preprint arXiv:2507.09882 (2025).
>
> [4] Yi, Ke, et al. "Learning topology-agnostic EEG representations with geometry-aware modeling." Advances in Neural Information Processing Systems 36 (2023): 53875-53891.

---

### Official Review · Reviewer_VgrX · 2025-10-31

**Soundness:** 2
**Presentation:** 1
**Contribution:** 1
**Rating:** 0
**Confidence:** 4

**Summary:**

This paper proposes MTSSRL-MD, a unified framework for multi-task self-supervised representation learning (SSRL) across multiple EEG datasets. The framework aims to improve generalization under label scarcity and heterogeneous EEG montages by integrating three components:

(1) Multi-dataset learning, to leverage diverse EEG sources and mitigate overfitting to specific datasets;

(2) Channel Alignment Module (CAM), a spatial attention mechanism that projects heterogeneous montages into a shared channel space;

(3) Multi-task SSL optimization, combining contrastive, temporal order, and spectral masking tasks with uncertainty-weighted loss balancing.

Experiments are conducted on three heterogeneous sleep-staging datasets (SleepEDF-20, ISRUC-S1, and ANPHY-Sleep) to validate the framework.

**Strengths:**

The research direction is valuable, addressing the problem of cross-dataset EEG representation learning, which is a crucial step toward robust and scalable EEG analysis.

**Weaknesses:**

**1. Unclear problem framing.**

While the paper’s objective is generally understandable, the introduction lacks a clear articulation of the core challenges motivating the use of multi-task learning and explicit montage modeling. It remains unclear what specific limitations the authors refer to when stating that “These strategies show the promise of dataset fusion but also its limitations.” The specific limitations of previous strategies should be explicitly defined, and the authors should further clarify why explicit montage alignment is necessary or beneficial, especially considering the potential inter-subject variability and the fact that implicit alignment methods may already capture spatial dependencies to some extent.

**2. Limited novelty.**

Each component of the proposed framework has clear prior art, and the main contribution lies in the integration of existing ideas rather than introducing substantial algorithmic novelty.

**3. Task scope limitation.**

Although framed as a unified framework for EEG representation learning, the experiments focus solely on sleep staging. Evaluating on additional EEG tasks (e.g., seizure detection, emotion recognition) would better support claims of generality.

**4. Unclear advantage of CAM.**

While CAM is central to the proposed framework, the paper does not quantitatively isolate its contribution or demonstrate why explicit alignment outperforms other approaches. In particular, recent foundation models for EEG representation learning, such as MMM and CBraMod, have already proposed more systematic solutions for montage alignment and domain generalization. The paper does not include comparisons with these representative methods and therefore fails to show a clear advantage or competitiveness in handling montage heterogeneity.

**5. Outdated baselines.**

The latest baseline referenced in this paper is from 2022 (ATCNet). For a fair and up-to-date evaluation, the authors should include comparisons with stronger and more recent sleep-staging baselines, as well as SSRL baselines (particularly EEG foundation models), published between 2023 and 2025.

**6. Minor presentation issues.**

- Line 112: “Self-supervised learning (SSRL)” should be “Self-supervised representation learning (SSRL)”.
- Section 3.4.1 is a single subheading and should be merged or renumbered.
- Page 8–9 reference jump issue.
- Code is not provided.

**Questions:**

1. Can the authors clearly define the core challenge and explain how each proposed component directly addresses it?

2. The three SSRL tasks are commonly used in self-supervised learning. The paper should clarify what advantages this particular combination offers and how their complementary effects are demonstrated.

3. How transferable is MTSSRL-MD to other EEG domains such as seizure detection or emotion classification?

4. Why choose uncertainty weighting over alternatives like GradNorm or PCGrad for task balancing?

5. Can the authors theoretically or empirically demonstrate why this kind of explicit montage alignment yields stronger generalization than others?

---

> ### Author Response · Authors · 2025-12-03
>
> Dear Reviewer VgrX,
>
> Thank you for the feedback!  We address your #1~#2 questions below.
> > Can the authors clearly define the core challenge and explain how each proposed component directly addresses it?
>
> Thank you for the question.
>
> Our work targets two core challenges in sleep EEG representation learning:
>
> 1. Cross-dataset montage heterogeneity
>
> Datasets use different channel sets (2, 6, 11 channels), electrode locations, and hardware, making direct dataset pooling infeasible.
>
> 2.  Limited and imbalanced labels
>
> Sleep-stage datasets provide few labeled epochs per subject and have strong class imbalance, making supervised training unstable.
>
> Each component of MTSSRL-MD is designed to address one side of this challenge:
>
> 1.	Channel Alignment Module (CAM) → addresses montage heterogeneity
> - CAM learns dataset-specific spatial projections that map heterogeneous montages into the same latent channel space. This removes the primary barrier that prevents multi-dataset pretraining.
> 2.	Shared encoder → leverages aligned inputs to learn cross-dataset generalizable structure
> - Once channels are aligned, the encoder can learn temporal–spectral patterns that are consistent across datasets, directly mitigating heterogeneity.
> 3.	Three SSL tasks → address label scarcity by providing richer supervision
> - Each task introduces a complementary inductive bias that compensates for missing labels: augmentation contrastive learning builds invariance to subject- and device-related variability, temporal shuffling captures stage-transition dynamics that supervised labels alone do not encode, and frequency-band masking learns sleep-relevant spectral patterns. Together, these tasks provide strong and diverse self-supervised signals that substantially reduce reliance of the model on labeled data.
> 4.	Uncertainty-weighted multi-task loss → stabilizes training across heterogeneous datasets and tasks
> - Different SSL tasks and datasets have different loss scales. Uncertainty weighting dynamically balances them, preventing instability and improving generalization under both dataset heterogeneity and label scarcity.
>
> CAM directly solves heterogeneous montages by combining the multi-task SSRL components and uncertainty weighting to tackle label scarcity. The unified framework enables effective multi-dataset and multi-task SSRL pretraining, which is not addressed by existing sleep EEG SSRL methods.
> > The three SSRL tasks are commonly used in self-supervised learning. The paper should clarify what advantages this particular combination offers and how their complementary effects are demonstrated.
>
> Thank you for the thoughtful question.
>
> Although augmentation contrast, temporal shuffling, and frequency-band masking are common SSRL task families, in our framework, they are not arbitrary choices. Each task is intentionally selected to capture a distinct and physiologically grounded property of sleep EEG, making the combination complementary rather than redundant.
>
> 1. Augmentation-Contrastive (AUG)
> - To promote invariance to inter-subject variability, sensor noise, and montage differences—crucial for heterogeneous EEG and cross-dataset robustness [1].
> 2. Temporal-Shuffling (TS)
> - To model temporal continuity and stage-transition dynamics, improving discrimination of ambiguous boundary epochs [2].
> 3. Frequency-Band Masking (FBM)
> - To encourage reliance on key spectral biomarkers, including N2 spindles, N3 delta power, and REM theta, which general SSRL objectives (e.g., MAE or contrastive-only losses) often underutilize [3].
>
> These tasks therefore target invariance, temporal dynamics, and spectral physiology—three complementary aspects that no single SSL objective can jointly capture. Our ablations (Table 4 in our manuscript) show that each task independently provides useful structure, but their combination yields the highest BACC and MF1 across all datasets.
>
> We have already revised Section 3.4.1 to more clearly explain this design choice and directly connect each task to the temporal, spectral, and invariance properties of sleep EEG.
>
> ***
> **Reference**
>
> [1] Mohsenvand, Mostafa Neo, Mohammad Rasool Izadi, and Pattie Maes. "Contrastive representation learning for electroencephalogram classification." Machine learning for health. PMLR, 2020.
>
> [2] Banville, Hubert, et al. "Uncovering the structure of clinical EEG signals with self-supervised learning." Journal of Neural Engineering 18.4 (2021): 046020.
>
> [3] Jo, Sangmin, et al. "Channel-aware self-supervised learning for eeg-based bci." 2023 11th International Winter Conference on Brain-Computer Interface (BCI). IEEE, 2023.

---

> ### Author Response · Authors · 2025-12-03
>
> Dear Reviewer VgrX,
>
> Thank you again for your time and efforts in reviewing our paper! We address your #3~#4 questions below.
> > How transferable is MTSSRL-MD to other EEG domains such as seizure detection or emotion classification?
>
> Thank you for the question.
>
> In this work, we intentionally focus on sleep staging because it is a domain that benefits most from multi-dataset and multi-task self-supervised learning. Sleep EEG datasets exhibit substantial cross-dataset variability in montage configurations, sampling rates, and acquisition hardware, making the Channel Alignment Module (CAM) particularly valuable.
>
> Moreover, sleep staging annotations are not only costly but also intrinsically difficult, with well-documented low inter-rater agreement. This combination of heterogeneous montages and challenging annotations makes sleep staging an ideal testbed for evaluating the effectiveness of our multi-dataset and label-efficient SSRL framework.
>
> MTSSRL-MD could be extended to other EEG domains, such as seizure detection or emotion recognition. However, transferring the framework to these tasks would require designing self-supervised objectives that capture the distinct temporal and event characteristics of those domains. As noted in Appendix I of the paper, we view this as promising future work, but it is beyond the scope of the present study, which focuses on improving downstream sleep staging performance.
>
> > Why choose uncertainty weighting over alternatives like GradNorm or PCGrad for task balancing?
>
> We thank the reviewer for the question.
>
> We choose uncertainty-weighted loss because it allows the model to learn task weights implicitly through homoscedastic uncertainties, without requiring manual tuning or additional optimization procedures, and has been widely adopted in EEG multi-task self-supervised learning [1, 2].
>
> In contrast, alternative balancing methods introduce non-trivial overhead:
>
> - GradNorm requires computing per-task gradient norms and tuning the additional hyperparameter α. Its objective is to equalize task training rates, which becomes unstable when task losses naturally differ in scale, as is the case in our three heterogeneous SSRL objectives. It also relies on specifying target loss ratios, which further increases the burden of hyperparameters in large-scale pretraining.
> - PCGrad performs pairwise gradient projections at every update step, requiring separate per-task gradients and substantially higher memory and compute overhead. While manageable for small task counts, this cost becomes prohibitive for large-scale pretraining across multiple datasets.
>
> Given our setting—multi-dataset, multi-task self-supervised pretraining with heterogeneous objectives—uncertainty weighting offers a clean, hyperparameter-light, and computationally efficient solution. It integrates naturally into our framework without introducing the additional cost of gradient normalization or per-task gradient projection.
> ***
> **Reference**
>
> [1] Li, Yang, et al. "GMSS: Graph-based multi-task self-supervised learning for EEG emotion recognition." IEEE Transactions on Affective Computing 14.3 (2022): 2512-2525.
>
> [2] Li, Guangqiang, et al. "MSLTE: multiple self-supervised learning tasks for enhancing EEG emotion recognition." Journal of Neural Engineering 21.2 (2024): 024003.

---

> ### Author Response · Authors · 2025-12-03
>
> Dear Reviewer VgrX,
>
> Thank you again for your time and efforts in reviewing our paper! We address your #5 question below.
> > Can the authors theoretically or empirically demonstrate why this kind of explicit montage alignment yields stronger generalization than others?
>
> Thank you for raising this important point.
>
> Cross-dataset EEG transfer is substantially influenced by heterogeneity in sensor layouts, channel counts, and acquisition setups, making spatial mismatch a well-recognized source of dataset shift in EEG analysis. This motivates the need for an explicit mechanism to harmonize heterogeneous montages before shared representation learning.
>
> Existing “implicit” strategies only partially address montage heterogeneity. Region-based pooling [1] normalizes channel dimensionality but reduces spatial resolution by merging electrodes. Graph-based methods [2] must build dataset-specific adjacency matrices, leaving montage differences as a persistent source of dataset shift. SeqCLR [3] avoids montage mismatch via single-channel inputs, but it necessarily discards multichannel spatial information that is crucial for spindles, K-complexes, slow waves, and REM dynamics.
>
> In contrast, our Channel Alignment Module (CAM) explicitly projects heterogeneous montages into a unified virtual channel space while preserving the multichannel structure. Empirically, CAM produces physiologically interpretable spatial weights, enabling MTSSRL-MD to consistently outperform the representative non-alignment SSRL baseline (SeqCLR) across datasets. As shown in Tables 1 and 2, CAM provides improvements of up to +12.42% in BACC and +11.37% in MF1 at 5% labels, and +9.30% (BACC) and +9.52% (MF1) at 10% fine-tuning labels.
>
> ***
> **Reference**
>
> [1] Tveitstøl, Thomas, et al. "Introducing Region Based Pooling for handling a varied number of EEG channels for deep learning models." Frontiers in Neuroinformatics 17 (2024): 1272791.
>
> [2] Han, Jinpei, Xiaoxi Wei, and A. Aldo Faisal. "EEG decoding for datasets with heterogenous electrode configurations using transfer learning graph neural networks." Journal of Neural Engineering 20.6 (2023): 066027.
>
> [3] Mohsenvand, Mostafa Neo, Mohammad Rasool Izadi, and Pattie Maes. "Contrastive representation learning for electroencephalogram classification." Machine learning for health. PMLR, 2020.

---

> ### Author Response · Authors · 2025-12-03
>
> Dear Reviewer VgrX,
>
> Thank you again for your time and efforts in reviewing our paper!
> > Line 112: “Self-supervised learning (SSRL)” should be “Self-supervised representation learning (SSRL)”.
>
> Thank you for the correction.
>
> We appreciate the reviewer’s attention to detail. We have already revised the terminology in Line 112 to “self-supervised representation learning (SSRL)” as suggested.
>
> > Section 3.4.1 is a single subheading and should be merged or renumbered.
>
> Thank you for pointing out the structural issue regarding the single subsection.
>
> To address this, we reorganized Section 3.4 by splitting it into two subsections:
>
> “Self-Supervised Representation Learning Tasks” and “Multi-Task Optimization (Uncertainty-Weighted)”, thereby removing the single-child heading and improving clarity.
>
> > Page 8–9 reference jump issue
>
> Thank you for pointing out this issue.
>
> The page break previously occurred within a citation due to LaTeX formatting. After revising the paragraph structure, the page break no longer intersects the citation, and the problem has now resolved in the revised manuscript.
>
>
> > Code is not provided
>
> Thank you for the comment.
>
> Following the conference guidelines, we selected to provide an anonymous link to our source code in the discussion forum.
> Source Code Link: https://github.com/mtssrl-md-eeg/MTSSRL-MD

---

### Official Review · Reviewer_8YwF · 2025-10-31

**Soundness:** 2
**Presentation:** 2
**Contribution:** 2
**Rating:** 4
**Confidence:** 3

**Summary:**

This paper proposes MTSSRL-MD, a framework for learning generalizable EEG representations across heterogeneous datasets for sleep stage classification. The method combines: (1) a Channel Alignment Module (CAM) to handle heterogeneous EEG montages, (2) three complementary self-supervised learning tasks (augmentation contrastive, temporal shuffling discrimination, and frequency band masking), and (3) uncertainty-weighted multi-task optimization. Experiments on three public sleep datasets (SleepEDF-20, ISRUC-S1, ANPHY-Sleep) demonstrate improvements over single-dataset SSRL and multi-dataset baseline SeqCLR, particularly in low-label regimes.

**Strengths:**

- The paper clearly articulates three critical challenges in EEG-based sleep staging: (a) label scarcity, (b) small dataset scale, and (c) montage heterogeneity.
- Experimental validation is quite comprehensive (multiple baselines, multiple metrics, different amounts of labels)
- Considerable improvements over SeqCLR and single-dataset SSRL baselines, while maintaining competitive performance at full supervision and achieving superior computational efficiency
- Results are supported through visualizations of CAM spatial attention

**Weaknesses:**

- Limited novelty in individual components of MTSSRL-MD (e.g., three SRRL tasks, CAM) The main contribution appears to be the engineering combination of existing techniques rather than fundamental algorithmic innovations.
- Limited amount of comparisons (SeqCLR is the main method and rather old), graph-based and feature fusion approaches are discussed but not empirically compared.
- Only three data sets are tested, all for sleep staging
- Ablation studies quite limited (only with respect to multi-task weighting strategies).

**Questions:**

- Can you provide a systematic ablation study that isolates the individual contributions of CAM and multi-task learning? Specifically, please report results for Basline, CAM-only, Multitask-only, full model.
- Why were recent multi-dataset EEG methods not included in the experimental comparison?  Comparing only against SeqCLR is insufficient to claim state-of-the-art performance.
- Have you evaluated MTSSRL-MD on any EEG tasks beyond sleep staging? E.g., seizure detection?
- When does MTSSRL-MD fail? Are there subjects or sleep patterns where the method struggles?
- Interpretability: Have you consulted with sleep clinicians or neurophysiologists to validate that the CAM spatial attention patterns (Fig 2) are clinically meaningful?
- How does MTSSRL-MD scale with increasing numbers of datasets? You tested 3 datasets - what happens with 5, 10, or 20 datasets?

---

> ### Author Response · Authors · 2025-12-03
>
> Dear Reviewer 8YwF,
>
> Thank you for the feedback!  We address your first question below.
> > Can you provide a systematic ablation study that isolates the individual contributions of CAM and multi-task learning? Specifically, please report results for Basline, CAM-only, Multitask-only, full model.
>
> Thank you for requesting a more systematic ablation. In the revised manuscript, we reorganize the results to align with the four configurations suggested by the reviewer and to separate the effects of CAM, multi-task learning, and multi-dataset pretraining.
>
> 1. Baseline: single SSRL task, no CAM.
>
> This setting corresponds to the single-task SSRL baselines (RP, TS, CPC) and the multi-dataset single-task method SeqCLR reported in Tables 1–3 (in our manuscript). All of these models utilize a single pretext task without CAM and therefore serve as our reference for evaluating the added value of CAM and multi-task learning.
>
> 2. CAM-only: CAM + single SSRL task (used to study single- vs multi-task).
>
> In Table 4 (in our manuscript), we fix the pretraining setup to multi-dataset + CAM and activate only one SSL task at a time (the first three rows, each with a single ✓ under AUG / TS / FBM). These rows implement CAM + single-task SSL and, together with the multi-task rows in the same table, allow us to isolate the effect of moving from single-task to multi-task SSL under a fixed CAM and multi-dataset configuration.
>
> 3. Multi-task-only: single-dataset + three SSRL tasks (used to study single vs multi-dataset).
>
> To isolate the effect of dataset diversity while keeping the multi-task setting fixed, Table 5 (in our manuscript) compares single-dataset pretraining (“S-only / I-only / A-only”) against multi-dataset pretraining with CAM (“S+I+A (CAM)”), all using the same three-task SSL objective and uncertainty weighting.  As shown in Table 5, multi-dataset pretraining consistently outperforms single-dataset pretraining across all three datasets, with clear gains in both BACC and MF1 under 5% and 10% label regimes.
>
> 4. Full MTSSRL-MD.
>
> The combination “multi-dataset + CAM + three SSL tasks with uncertainty weighting” (last row of Table 4 and the “S+I+A (CAM)” rows in Table 5) corresponds to our full model. Tables 1–3 further show that this full configuration outperforms all supervised and SSRL baselines on every dataset and label regime.
>
> We intentionally present the ablations in two complementary tables. Table 4 for the task-level axis (single-task vs multi-task under fixed multi-dataset + CAM) and Table 5 for the dataset-level axis (single-dataset vs multi-dataset + CAM under fixed multi-task). Merging everything into a single large table would mix task and dataset variations, making it harder to interpret the contribution of each module.
>
> The results consistently show that CAM contributes to spatial alignment, multi-task learning improves temporal–spectral robustness, and multi-dataset pretraining provides additional gains.

---

> ### Author Response · Authors · 2025-12-03
>
> Dear Reviewer 8YwF,
>
> We sincerely appreciate your thoughtful comments. We address your #2~#3 questions below.
>
> > Why were recent multi-dataset EEG methods not included in the experimental comparison? Comparing only against SeqCLR is insufficient to claim state-of-the-art performance.
>
> We appreciate the opportunity to clarify the scope of our literature review. We have reviewed the EEG pretraining literature from 2023 to 2025 and updated the Related Work section accordingly in our revised manuscript.
> These recent developments fall mainly into two research directions:
>
> 1. Large-scale foundation models for universal EEG representation learning.
>
> The works such as LaBraM [1], CBraMod [2], and AdaBrain-Bench [3] focus on constructing general-purpose EEG models trained on millions of signals across numerous datasets and tasks. These frameworks introduce valuable insights for broad EEG representation learning. However, they assume large-scale, heterogeneous corpora, and are typically optimized for general-purpose transfer, rather than specialized, low-channel sleep EEG. Their fine-tuning requires substantial computational resources and large labeled datasets, which contradicts our problem setup.
>
> 2. Geometry-aware or topology-informed EEG modeling.
>
> The works, such as Yi et al. [4], rely on stable electrode geometry and dense montages to encode spatial structure. These settings are not appropriate for the low-density and heterogeneous sleep-EEG configurations used in our setting. (SleepEDF-20 provides only two EEG channels, while three datasets in our study use non-overlapping, low-density montages)
>
> While these lines of research are important advances, they target different problem formulations—either large-scale universal modeling or geometry-aware spatial SSL—rather than label-efficient cross-dataset sleep staging with low-density EEG. Nonetheless, we agree that acknowledging these developments improves the contextualization of our contribution. We have already included a dedicated discussion of these recent methods in the Related Work section of our revised manuscript, which clarifies their relevance to our problem setting.
> >Have you evaluated MTSSRL-MD on any EEG tasks beyond sleep staging? E.g., seizure detection?
>
> Thank you for the question!
>
> In this work, we intentionally focus on sleep staging because it is a domain that benefits most from multi-dataset and multi-task self-supervised learning. Sleep EEG datasets exhibit substantial cross-dataset variability in montage configurations, sampling rates, and acquisition hardware, making the Channel Alignment Module (CAM) particularly valuable.
> Moreover, sleep staging annotations are not only costly but also intrinsically difficult, with well-documented low inter-rater agreement. This combination of heterogeneous montages and challenging annotations makes sleep staging an ideal testbed for evaluating the effectiveness of our multi-dataset and label-efficient SSRL framework.
>
> MTSSRL-MD could be extended to other EEG domains, such as seizure detection or emotion recognition. However, transferring the framework to these tasks would require designing self-supervised objectives that capture the distinct temporal and event characteristics of those domains.
>
> As noted in Appendix I of the paper, we view this as promising future work, but it is beyond the scope of the present study, which focuses on improving downstream sleep staging performance.
>
> ***
> **Reference**
>
> [1] Jiang, Wei-Bang, Li-Ming Zhao, and Bao-Liang Lu. "Large brain model for learning generic representations with tremendous EEG data in BCI." arXiv preprint arXiv:2405.18765 (2024).
>
> [2] Wang, Jiquan, et al. "Cbramod: A criss-cross brain foundation model for eeg decoding." arXiv preprint arXiv:2412.07236 (2024).
>
> [3] Wu, Jiamin, et al. "Adabrain-bench: Benchmarking brain foundation models for brain-computer interface applications." arXiv preprint arXiv:2507.09882 (2025).
>
> [4] Yi, Ke, et al. "Learning topology-agnostic EEG representations with geometry-aware modeling." Advances in Neural Information Processing Systems 36 (2023): 53875-53891.

---

> ### Author Response · Authors · 2025-12-03
>
> Dear Reviewer 8YwF,
>
> Thank you again for your time and efforts in reviewing our paper! We address your #4~#5 questions below.
> >When does MTSSRL-MD fail? Are there subjects or sleep patterns where the method struggles?
>
> Thank you for raising this important point.
>
> Understanding failure modes is indeed essential for assessing clinical applicability. In our experiments, the primary failure mode of MTSSRL-MD occurs in the N1 stage, which is a well-known minority class in sleep staging.
> SleepEDF-20 is highly imbalanced, with N1 accounting for only 2.64% of all epochs (Table 6 in manuscript). This severe under-representation makes N1 intrinsically difficult even for supervised models, and the difficulty is amplified under low-label fine-tuning (5–10%), where only a few labeled N1 samples are available.
> Consistent with this, our class-wise results (Table 5 in manuscript) show that N1 receives the lowest F1-score (34.06%), with errors concentrated in W–N1 and N1–N2 transition epochs—regions documented to exhibit the lowest inter-rater agreement [1,2]. Even under this challenging minority-class setting, MTSSRL-MD achieves 34.06% F1 on N1, outperforming the multi-dataset SeqCLR baseline (32.07%) and clearly exceeding single-dataset SSL methods (21.26–28.52%).
>
> >Interpretability: Have you consulted with sleep clinicians or neurophysiologists to validate that the CAM spatial attention patterns (Fig 2) are clinically meaningful?
>
> Thank you for raising this important point. While we did not conduct formal interviews with sleep clinicians or neurophysiologists, we performed a detailed interpretability analysis of the Channel Alignment Module (CAM), and the results are reported in Appendix G.
>
> Our analysis shows that the spatial attention patterns learned by CAM closely align with established findings in sleep neurophysiology across all three datasets. In SleepEDF-20, which contains only two channels, CAM assigns nearly identical weights to Fpz and Pz, indicating that it preserves all available information in sparse montages—consistent with evidence that key sleep dynamics, including slow-wave activity, can be captured from midline and fronto-parietal derivations [3]. In ISRUC-S1, CAM assigns higher weights to F3 and F4, aligning with the known fronto-central distribution of REM-related sawtooth waves reported in prior studies [4]. For ANPHY-Sleep, which provides high-density EEG, CAM emphasizes O1 and F4, followed by C3 and O2/Pz. This pattern coheres with the established generators of major sleep rhythms: occipital sites contributing to posterior alpha [5], fronto-central sites to N3 slow waves [6], N2 sleep spindles [7], and REM sawtooth activity [4], with activity extending toward midline parietal regions such as Pz [8]. Together, these dataset-specific patterns indicate that CAM consistently highlights physiologically meaningful regions rather than arbitrary channels, leveraging complementary spatial EEG sources when aligning heterogeneous montages.
>
> Although clinical expert validation is beyond the scope of this work, we agree that incorporating expert assessment is a valuable direction for future research. We have already revised Appendix G to clarify where these analyses are presented and to highlight their physiological basis better.
>
> ***
> **Reference**
>
> [1] Stephansen, Jens B., et al. "Neural network analysis of sleep stages enables efficient diagnosis of narcolepsy." Nature communications 9.1 (2018): 5229.
>
> [2] Lee, Yun Ji, et al. "Interrater reliability of sleep stage scoring: a meta-analysis." Journal of Clinical Sleep Medicine 18.1 (2022): 193-202.
>
> [3] Kemp, Bob, et al. "Analysis of a sleep-dependent neuronal feedback loop: the slow-wave microcontinuity of the EEG."IEEE Transactions on Biomedical Engineering 47.9 (2000): 1185-1194.
>
> [4] Frauscher, Birgit, et al. "Rapid eye movement sleep sawtooth waves are associated with widespread cortical activations." Journal of Neuroscience 40.46 (2020): 8900-8912.
>
> [5] Halgren, Mila, et al. "The generation and propagation of the human alpha rhythm." Proceedings of the National Academy of Sciences 116.47 (2019): 23772-23782.
>
> [6] Massimini, Marcello, et al. "The sleep slow oscillation as a traveling wave." Journal of Neuroscience 24.31 (2004): 6862-6870.
>
> [7] De Gennaro, Luigi, and Michele Ferrara. "Sleep spindles: an overview." Sleep medicine reviews 7.5 (2003): 423-440.
>
> [8] Nir, Yuval, et al. "Regional slow waves and spindles in human sleep." Neuron 70.1 (2011): 153-169.

---

> ### Author Response · Authors · 2025-12-03
>
> Dear Reviewer 8YwF,
>
> Thank you for the feedback! We address your #6 question below.
> > How does MTSSRL-MD scale with increasing numbers of datasets? You tested 3 datasets - what happens with 5, 10, or 20 datasets?
>
> Thank you for the insightful question.
>
> Although we evaluated only three datasets, our additional ablations show that multi-dataset + multi-task SSL consistently outperforms single-dataset SSL, demonstrating that MTSSRL-MD directly benefits from increased dataset diversity (Table 5 in revised manuscript). This supports the conclusion that scaling to more datasets strengthens the learned representation.
>
> From an architectural perspective, MTSSRL-MD demonstrates strong scalability: CAM aligns heterogeneous montages with constant parameter cost, and the SSL tasks benefit from broader temporal–spectral variability.
>
> Thus, we expect performance to continue improving with 5, 10, or more datasets, and we plan to include such results in future work.

---

### Official Review · Reviewer_R97R · 2025-10-31

**Soundness:** 1
**Presentation:** 2
**Contribution:** 1
**Rating:** 2
**Confidence:** 4

**Summary:**

The paper proposes a self-supervised learning framework for EEG for a multi-dataset, multi-task context. It uses a Channel Alignment Module to handle different EEG montages. While the paper is clearly written and addresses the important problem of cross-dataset generalization, its contributions are significantly undermined by a severe lack of relevant baseline comparisons and ablation studies being absent for one of its primary methodological proposals.

**Strengths:**

- The paper is well-written and clearly articulates its methods and aims. The goal of creating a unified SSL framework that is cross-domain is important to the field.
- The combination of multiple pretext tasks which have been shown to perform well is a solid choice; combining this with the uncertainty weighting of these multiple objectives is very sensible and is shown to improve effectiveness.

**Weaknesses:**

- **Severe lack of baseline comparisons**: A primary claim of the paper is the benefit of the multi-dataset approach, yet the paper does not compare against any recent multi-dataset SSL methods for EEG. Examples would have included EEGPT, BIOT, Labram, CBramod, among others. A single comparison to a workshop paper from 2020 (SeqCLR) is simply insufficient to judge the method's performance relative to the literature. Similarly for single-dataset baselines; These comparisons are also insufficient, relying on methods from 2021. To make the current paper relevant, it would need to either show that a combination of methods by Banville et al. (2021) is at least as good as current methods, or provide insight into how using and weighting multiple pretext tasks interacts with current/modern pretext strategies.

- **No ablation or analysis on the Channel Alignment Module**: A central component of the paper is the Channel Alignment Module yet no ablation studies are performed to validate its effectiveness. Although it aims to address precisely the same issue other cross-domain methods tackle (e.g. tokenizers in Labram or Cbramod), no comparisons either theoretical or empirical are provided.

- **Limited methodological novelty and/or insight**: The paper largely combines existing pieces (known pretext tasks, uncertainty weighting). The only potentially novel piece is the CAM, which is unverified. As the paper stands, its only clear contribution is showing that multi-task pretraining is useful, which has now been shown in multiple publications. The work fails to provide new and interesting insights for the field and feels incomplete as is.

- **Conflation of contributions**: The paper appears to marry the concepts of pretraining tasks and single vs multi-dataset pretraining. These are fairly independent from my understanding; SeqCLR can be performed on single datasets and singular Banville et al. methods can be applied in a multi-dataset setting. The paper would be stronger if it decoupled these contributions and analyzed them independently.


In its current state, the paper feels seriously incomplete. The lack of crucial baseline comparisons and any ablation or analysis of the proposed Channel Alignment Module makes it impossible to validate the paper's claims or understand its relevance. The paper does not offer sufficient new insight to warrant acceptance.

**Questions:**

1. To make the paper more relevant to the field, are you able to present analyses for pretraining with modern methods? (Either as comparisons or added in during multi-task pretraining)

2. Did you perform any analyses into the channel alignment module?

3. Do the authors see any problems with disentangling the pretext tasks from single- vs multi-dataset training? If not, why was this not done?

Minor:
- Figure 2 Presentation: Figure 2 appears to be raw matplotlib output with very small fonts, making it difficult to read. Here the presentation could be improved.

---

> ### Author Response · Authors · 2025-12-03
>
> Dear Reviewer R97R,
>
> Thank you for the feedback!  We address your comments below.
>
> > To make the paper more relevant to the field, are you able to present analyses for pretraining with modern methods? (Either as comparisons or added in during multi-task pretraining)
>
> We appreciate the opportunity to clarify the scope of our literature review. We have reviewed the EEG pretraining literature from 2023 to 2025 and updated the Related Work section accordingly in our revised manuscript.
> These recent developments fall mainly into two research directions:
>
> 1. Large-scale foundation models for universal EEG representation learning.
>
> The works such as LaBraM [1], CBraMod [2], and AdaBrain-Bench [3] focus on constructing general-purpose EEG models trained on millions of signals across numerous datasets and tasks. These frameworks introduce valuable insights for broad EEG representation learning. However, they assume large-scale, heterogeneous corpora, and are typically optimized for general-purpose transfer, rather than specialized, low-channel sleep EEG. Their fine-tuning requires substantial computational resources and large labeled datasets, which contradicts our problem setup.
>
> 2. Geometry-aware or topology-informed EEG modeling.
>
> The works, such as Yi et al. [4], rely on stable electrode geometry and dense montages to encode spatial structure. These settings are not appropriate for the low-density and heterogeneous sleep-EEG configurations used in our setting. (SleepEDF-20 provides only two EEG channels, while three datasets in our study use non-overlapping, low-density montages)
>
> While these lines of research are important advances, they target different problem formulations—either large-scale universal modeling or geometry-aware spatial SSL—rather than label-efficient cross-dataset sleep staging with low-density EEG. Nonetheless, we agree that acknowledging these developments improves the contextualization of our contribution. We have already included a dedicated discussion of these recent methods in the Related Work section of our revised manuscript, which clarifies their relevance to our problem setting.
>
> > Did you perform any analyses into the channel alignment module?
>
> Thank you for the question!
>
> Yes, we performed detailed analyses of the Channel Alignment Module, and the results are reported in Appendix G.
> 1. Hyperparameter Analysis — Appendix G.1
>
> We evaluated sensitivity of CAM to its two key design choices (positional embedding dimension and number of virtual channels).
> Across all three datasets, the heatmaps in Figs. 4–5 show stable trends, confirming that our chosen configuration (PositionDim = 72, CM = 16) achieves consistently strong BACC/MF1 under heterogeneous montage conditions.
>
> 2. CAM Weight Interpretability — Appendix G.2
>
> Fig. 2 illustrates spatial attention weights learned by CAM across the three datasets. These patterns suggest that CAM adapts to different montage configurations, tends to reduce redundancy in high-density settings, retains critical information in sparse ones, and highlights channels that align with established sleep biomarkers, thereby supporting both effective alignment and physiologically interpretable weighting.
>
> > Do the authors see any problems with disentangling the pretext tasks from single- vs multi-dataset training? If not, why was this not done?
>
> Thank you for the question!
>
> We agree that it is crucial to verify whether the improvements in MTSSRL-MD arise from the multi-task objectives themselves or from the additional dataset diversity.
>
> To isolate this factor, we added a new ablation that keeps the pretext tasks fixed (the same multi-task SSRL configuration) and varies only the dataset setting—single-dataset vs. multi-dataset pretraining.
>
> As shown in the updated Table 5 (in revised manuscript), multi-dataset pretraining consistently outperforms single-dataset pretraining across all datasets, with gains of +2.97 to +5.75 BACC and +3.79 to +7.35 MF1 relative improvement under the 5% label regime and +3.57 to +5.28 BACC and +3.43 to +6.41 MF1 under the 10% label regime.
>
> This confirms that the improvements do not stem solely from the SSRL task design; dataset diversity contributes substantially and directly to the performance gains. We have included this ablation in the revised manuscript.
>
>
> ***
> **Reference**
>
> [1] Jiang, Wei-Bang, Li-Ming Zhao, and Bao-Liang Lu. "Large brain model for learning generic representations with tremendous EEG data in BCI." arXiv preprint arXiv:2405.18765 (2024).
>
> [2] Wang, Jiquan, et al. "Cbramod: A criss-cross brain foundation model for eeg decoding." arXiv preprint arXiv:2412.07236 (2024).
>
> [3] Wu, Jiamin, et al. "Adabrain-bench: Benchmarking brain foundation models for brain-computer interface applications." arXiv preprint arXiv:2507.09882 (2025).
>
> [4] Yi, Ke, et al. "Learning topology-agnostic EEG representations with geometry-aware modeling." Advances in Neural Information Processing Systems 36 (2023): 53875-53891.

---

> ### Author Response · Authors · 2025-12-03
>
> Dear Reviewer R97R,
>
> > Figure 2 Presentation: Figure 2 appears to be raw matplotlib output with very small fonts, making it difficult to read. Here the presentation could be improved.
>
> Thank you for pointing this out!
>
> We appreciate the reviewer’s suggestion regarding the readability of Figure 2. In the revised manuscript, we have already increased the font sizes of axis labels, tick labels, and numerical annotations, and improve the overall figure formatting to ensure the visualization is clearer and easier to interpret.

---

### Author Response · Authors · 2025-12-03
**Code Repository**

Following the conference guidelines, we selected to provide an anonymous link to our source code in the discussion forum.
Source Code Link: https://github.com/mtssrl-md-eeg/MTSSRL-MD

---

### Meta-Review · Area_Chair_zjoP · 2026-01-03

**Summary:**

The reviewers acknowledged the paper's solid engineering, clear presentation, and relevance to label-efficient EEG learning. However, several remained concerns limited overall enthusiasm, as listed as below.
* Outdated and insufficient baselines: Lack of direct comparison with recent EEG foundation or large-scale pretraining models (All the reviewers).
* Limited methodological novelty: The main components are viewed as integration of existing techniques rather than a new conceptual advance (Reviewers R97R, 8YwF, VgrX, z6TT).
* Narrow task scope: Evaluation largely limited to sleep staging, raising concerns about generalization to other tasks upon the EEG signals (Reviewers 8YwF, VgrX).
* Questionable generalization of CAM: CAM has not been well verified (Reviewers R97R, VgrX) and may be dataset or montage specific (Reviewer LS6M).

These concerns collectively led reviewers to view the contribution as incremental. Most concerns have not been well addressed. I'm not recommending this paper.

**Reviewer Concerns:**

The concerns that have been addressed:
* CAM effectiveness and interpretability
* Confounding between multi-task and multi-dataset learning
* Training protocol clarity and presentation issues
* Justification for uncertainty-weighted loss

The remained concerns that are still outstanding:
* Lack of direct experimental comparison with 2023–2025 EEG foundation models
* Limited methodological novelty beyond integration of known components
* Narrow task domain (sleep staging only)
* Generalization claims constrained by dataset-aware CAM design

**Reviewer Scores:**

Based on the tone and content of the reviews, none of the reviewers would be expected to change their original scores after the discussion.

---

### Decision · Program_Chairs · 2026-01-26

Reject